# Evaluating L-band InSAR Snow Water Equivalent Retrievals with Repeat Ground-Penetrating Radar and Terrestrial Lidar Surveys in Northern Colorado

Randall Bonnell[1], Daniel McGrath[1], Jack Tarricone[2,3], Hans-Peter Marshall[4], Ella Bump[5], Caroline Duncan[6], Stephanie Kampf[5], Yunling Lou[7], Alex Olsen-Mikitowicz[5], Megan Sears[5], Keith Williams[8], Lucas Zeller[1], and Yang Zheng[7]

[1]Department of Geosciences, Colorado State University, Fort Collins, Colorado, USA
[2]Hydrological Sciences Laboratory, NASA Goddard Space Flight Center, Greenbelt, Maryland, USA
[3]NASA Postdoctoral Program, NASA Goddard Space Flight Center, Greenbelt, Maryland, USA
[4]Department of Geosciences, Boise State University, Boise, Idaho, USA
[5]Department of Ecosystem Science and Sustainability, Colorado State University, Fort Collins, Colorado, USA
[6]Alaska District, U.S. Army Corps of Engineers, Anchorage, Alaska, USA
[7]Jet Propulsion Laboratory, California Institute of Technology, Pasadena, California, USA
[8]GAGE Facility, UNAVCO Inc., Boulder, Colorado, USA

*Correspondence to*: Randall Bonnell (randall.bonnell@colostate.edu)

**Abstract.** Snow provides critical water resources for billions of people, making the remote sensing of snow water equivalent (SWE) a highly prioritized endeavor, particularly given ongoing climate change impacts. Synthetic Aperture Radar (SAR) is a promising method for remote sensing of SWE because radar penetrates snow, and SAR interferometry (InSAR) can be used to estimate changes in SWE ($\Delta$SWE) between SAR acquisitions. We calculated $\Delta$SWE retrievals from 10 NASA L-band (1–2 GHz, ~25 cm wavelength) Uninhabited Aerial Vehicle SAR (UAVSAR) acquisitions covering a ~640 km$^2$ swath in northern Colorado during the winters of 2020 and 2021. UAVSAR acquisitions coincided with ~117 mm of accumulation in 2020 and ~282 mm of accumulation in 2021. $\Delta$SWE retrievals were evaluated against measurements of SWE from repeat ground-penetrating radar (GPR) and terrestrial lidar scans (TLS) collected during the NASA SnowEx Time Series Campaigns at two field sites (total area = ~0.2 km$^2$) as well as SWE measurements from seven automated stations distributed throughout the UAVSAR swath. For single InSAR pairs, UAVSAR $\Delta$SWE retrievals yielded an overall r of 0.72–0.79 and RMSE of 19–22 mm when compared with TLS and GPR $\Delta$SWE retrievals. UAVSAR $\Delta$SWE showed some scatter with $\Delta$SWE measured at automated stations for both study years, but cumulative UAVSAR SWE yielded a r = 0.92 and RMSE = 42 mm when compared to total SWE measured by the stations. Further, UAVSAR $\Delta$SWE RMSEs differed by <10 mm for coherences (i.e., the complex interferometric coherence) of 0.10 to 0.90, suggesting that coherence has only a small influence on the $\Delta$SWE retrieval accuracy. Given the evaluations presented here and in other recent studies, the upcoming NASA-ISRO SAR (NISAR) satellite mission, with a 12-day revisit period, offers an exciting opportunity to apply this methodology globally.

# 1 Introduction

In snow-dominated watersheds, melt from seasonal snow drives streamflow and groundwater recharge (Li et al., 2017; Lorenzi et al., 2024). Globally, snowmelt supplies water resources for more than one-sixth of the population (Barnett et al., 2005). However, warming temperatures are decreasing the probability of snowfall in historically snow-dominated watersheds (Klos et al., 2014; McCrystall et al., 2021), shifting snowpacks to higher elevations and more poleward latitudes, and effectively decreasing the predictability of streamflow in these basins (Siirila-Woodburn et al., 2021). Mountains store a disproportionately large amount of snow despite comprising a small fraction of the global land surface (Wrzesien et al., 2019). Yet, in the mountains of the western United States, climate change has driven a 15–30% decline in snow water equivalent (SWE), the defining snowpack hydrologic variable, and SWE is expected to decline by an additional 25% by 2050 (Mote et al., 2018; Siirila-Woodburn et al., 2021). The projected changes are acute globally; by 2100, snowmelt is projected to decline in the European Alps by 50% (Moraga et al., 2021), while basins in the Himalayas may see snowfall declines of 30–70% for the warmest climate scenarios (Viste and Sorteberg, 2015). Although snowpack monitoring via automated stations exists in some countries (e.g., SNOTEL stations in the United States; Fleming et al., 2023), location bias, limited elevational range, and large spatial variability in snow over short length scales results in an incomplete characterization of this resource (Dozier et al., 2016). Thus, satellite remote sensing methods for snowpack monitoring at

high resolution (<500 m, <weekly) have been set as a high priority for study and development by the National Academies of Sciences, Engineering, and Medicine (National Academies of Sciences, Engineering, and Medicine, 2018).

The remote sensing of SWE is challenged by environmental factors (i.e., topography, vegetation) and by the spatiotemporally varying physical parameters of the snowpack (i.e., SWE, density, liquid water content, snow grain size). The NASA SnowEx Campaigns were conducted from 2017–2023 in the western United States to evaluate and develop remote sensing methods for snowpack monitoring, with the retrieval of SWE set as a primary goal (Durand et al., 2018). SWE is calculated as the product of snow depth and snow density, and there are two primary groups of techniques for remote

sensing of SWE at high spatial resolutions (<500 m): i) depth-based optical-infrared methods and ii) radar-based methods. Depth-based optical-infrared methods (e.g., stereo satellite imagery, lidar) require cloud-free conditions and derive snow depths by differencing a snow-off digital elevation model (DEM) from a snow-on DEM (Currier et al., 2019; Hu et al., 2023). Snow density model estimates or in situ measurements are required to convert the snow depths to SWE (e.g., Hedrick et al., 2018), which adds to the uncertainty of this technique (Raleigh and Small, 2017). Both satellite lidar (e.g., Besso et al.,

2024) and very-high resolution stereo satellite imagery (e.g., Hu et al., 2023) are being explored as depth-based methods for the remote sensing of SWE. Radar approaches are distinct from depth-based approaches because the radar signal penetrates the snowpack. Satellite radar approaches for snow depth and SWE retrievals are implemented from synthetic aperture radar (SAR) platforms and the techniques for snow depth and SWE remote sensing are primarily grouped into backscatter approaches, which use the amplitude component of the radar signal to derive snow depth and/or SWE, and time-of-flight

approaches, which derive SWE from the signal path length and includes SAR interferometry (InSAR). A third approach, which uses the co-polar phase difference, has also been tested. Readers interested in the co-polar phase difference methodology are referred to Leinss et al. (2014) and Patil et al. (2020).

    Unlike optical-infrared methods, SAR approaches for snow remote sensing are not limited by cloud cover, primarily due to low atmospheric absorption at radar frequencies (Woodhouse, 2017). For SAR backscatter approaches, the radar signal is

transmitted through the snowpack, and the signal is backscattered to the sensor via volume scattering from snow grains and rough scattering from the snow-ground interface (Tsang et al., 2022). Early backscatter work found that combined X- (8–12 GHz, ~3 cm wavelength) and C-band (4–8 GHz, ~5 cm wavelength) SAR was capable of measuring snow depths from 0.5– 2.5 m (RMSE = 0.34 m; Shi and Dozier, 2000). More recent efforts have emphasized combined X- and Ku-band (12–18 GHz, ~1.8 cm wavelength) SAR; these backscatter approaches are promising methods for measuring SWE in shallow

snowpacks (<150 mm; Tsang et al., 2022), with the potential for retrieving SWE in deeper conditions (Borah et al., 2023). C-band backscatter approaches are capable of measuring snow depths in deeper snowpacks (>1 m), albeit with higher uncertainty (Lievens et al., 2019, 2022). Backscatter approaches have known uncertainties in wet snow conditions, at large incidence angles, and in forests (Lievens et al., 2022; Tsang et al., 2022). InSAR is a unique method for retrieving SWE because the interferometric phase change has a near linear relation to SWE change (Guneriussen et al., 2001). In dry snow,

this characteristic can be used to retrieve changes in SWE without a priori information on snowpack density with an estimated 7% uncertainty related to the linear approximation (Leinss et al., 2015). Applying the InSAR technique at low-

frequency (e.g., L-band, ~25 cm wavelength) limits interaction between the radar signal and snow grains, increases the signal penetration in forests and in wet snow (Naderpour et al., 2022), and promotes increased coherence (described below) over longer temporal baselines (Ruiz et al., 2022). A review of the transmissibility of L-band radar in snow is provided in

Appendix A.1. With the upcoming launches of L-band SAR satellites, such as the NASA-ISRO SAR satellite (NISAR), the Radar Observing System for Europe satellite (ROSE-L), and the Tandem-L Interferometric Radar Mission, radar products will be publicly available at high spatial and temporal resolution across the globe (NISAR: 80 m spatial resolution, 12-day repeat; ISRO Space Applications Centre, 2023).

InSAR is a change detection method that measures the phase change between repeat SAR acquisitions and relies upon a

coherent reflection from the snow-ground interface (Guneriussen et al., 2001; Appendix A.2). The InSAR SWE retrieval technique was first established at C-band from the European Remote-Sensing Satellite (ERS) platform at a field site in Norway. The study showed that snowfall could be mistaken as a deformation signal in interferograms (the interferometric phase change data product; Guneriussen et al., 2001). Deeb et al. (2011) applied this technique to the ERS satellite using repeat acquisitions during an accumulation season to measure changes in SWE (ΔSWE) at a site on the North Slope of

Alaska, United States that revealed ΔSWE spatial patterns correlated with the prevailing wind direction. Since then, the technique has been tested for multi-year, season-long ΔSWE retrievals from a tower mounted platform in Finland at Ku-, X-, C-, and L-band frequencies (Leinss et al., 2015; Ruiz et al., 2022), by several studies emphasizing one or two interferometric pairs (Conde et al., 2019; Marshall et al., 2021; Nagler et al., 2022; Palomaki and Sproles, 2023; Tarricone et al., 2023), and by two season-long studies that used a time series of interferometric pairs (Hoppinen et al., 2024; Oveisgharan et al., 2024).

In general, these studies have found that InSAR ΔSWE retrievals are highly correlated with in situ measurements, but accuracy has varied on a case-by-case basis and in situ measurements for validation have been limited to point-based measurements that likely do not capture the spatial complexity of the snowpack. Additionally, only three of these studies have considered atmospheric signal delays, which represent a source of uncertainty because changes in atmospheric pressure and water content can further affect the ΔSWE retrieval accuracy (Gong et al., 2013).

Here, coherence refers to the complex interferometric coherence and is a measure of the similarity of the backscattered radar signal properties between two acquisitions (Woodhouse, 2017). Coherence is considered an index for confidence in phase change measurements, where phase changes with higher coherences are considered more accurate, and coherence must be maintained for the accurate unwrapping of interferograms. Coherence is affected by forest cover, changes in soil conditions (e.g., soil moisture changes or freeze-thaw changes), changes in the dielectric permittivity of the snowpack (e.g.,

melt-refreeze cycles), snow metamorphism (Brangers et al., 2023), and significant snow accumulation/ablation events (Ruiz et al., 2022). Collectively, these factors indicate that as the temporal baseline (i.e., time interval) between interferometric pairs is extended and/or major snowpack changes occur, coherence will degrade (Deeb et al., 2011), particularly at higher frequencies (Ruiz et al., 2022). A review of the calculation of coherence is provided in Appendix A.2.

Here, we calculated ΔSWE retrievals from 10 L-band NASA Uninhabited Aerial Vehicle SAR (UAVSAR; Rosen et al.,

2006) InSAR pairs collected during the NASA SnowEx Time Series campaigns in 2020 and 2021 over north-central

Colorado. During UAVSAR acquisitions, we collected spatially distributed ground-penetrating radar (GPR) at a very similar frequency to UAVSAR (UAVSAR = 1.26 GHz, GPR = 1.0 GHz) for all InSAR pairs, and we performed terrestrial lidar scans (TLS) for two InSAR pairs. Our study examines three components of InSAR ΔSWE retrievals. First, we leveraged our ground observations to evaluate the accuracy of the L-band InSAR technique for ΔSWE retrievals for two accumulation
seasons in a dry continental subalpine snowpack. We then evaluated UAVSAR ΔSWE retrieval errors against coherence to examine it as a potential metric for ΔSWE retrieval accuracy. Finally, UAVSAR ΔSWE retrievals are summed across each individual winter season and compared with total SWE measured at seven automated stations to evaluate the accuracy of the technique across a time series.

## 2 Overview of SnowEx 2020 and 2021 at Cameron Pass, Colorado

The SnowEx 2020 Time Series campaign was originally planned for a single season at 13 field sites (Marshall et al., 2019), but was cut short due to the COVID-19 pandemic and subsequently restarted in 2021 at seven field sites. Weekly to bi-weekly surveys were performed at Cameron Pass, Colorado (Figure 1a), coinciding with UAVSAR flights (Table 1). The flight line was typically ~40 km in length with a swath width of 16 km, but deviations from the spatial baseline and poor GNSS accuracy caused data acquisitions to be shortened for a few dates. The primary flight heading was southeast (141°),
with a secondary northwest heading (321°) flown when time allowed. For the analysis, we used the 141° heading for all InSAR pairs except the 27 January to 3 February 2021 interval, which used the 321° heading.

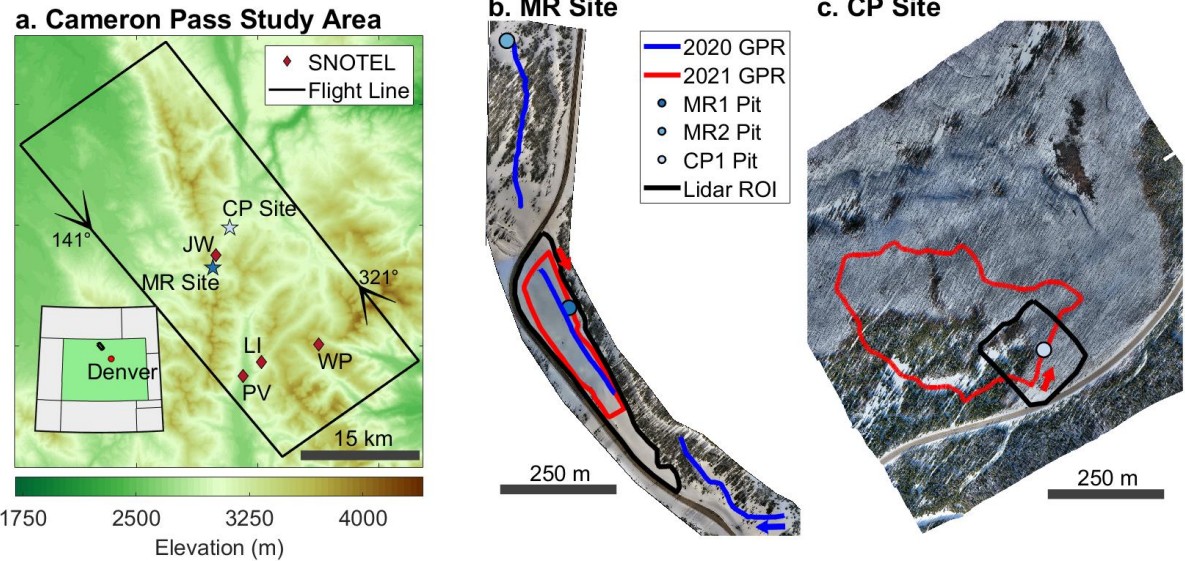

**Figure 1:** (a) Cameron Pass study area showing the Rocky Mountains, CO UAVSAR flight line overlaid on the Copernicus DEM (European Space Agency, 2021) with flight headings indicated by arrows. Locations are given for the Michigan River (MR) field site,
Cameron Peak (CP) field site, and the Joe Wright (JW), Willow Park (WP), Lake Irene (LI), and Phantom Valley (PV) SNOTEL stations.

Inset depicts the location of the flight line in Colorado. Middle and right panels show uncrewed aerial vehicle (UAV) imagery collected during March 2020 at the (b) MR field site and February 2021 at the (c) CP field site. The MR field site was surveyed during 2020 and 2021, whereas the CP field site was only surveyed during 2021. Key study areas, including snow pit locations, GPR transects, and terrestrial lidar regions of interest (Lidar ROI) are plotted. Arrows indicate the starting location and travel direction of the GPR transects.

**Table 1:** UAVSAR flight dates and times, field survey dates, GPR survey times, and ground observations performed for each field survey date. For instances where both the 141° and 321° flight headings were used, flight times are given for both. Otherwise, only flight times for the 141° heading are listed. For 2021, GPR survey times are given for the Michigan River (MR) and the Cameron Peak (CP) field sites. Ground observations include GPR, TLS, snow pits (SP), and probed depths (PD).

| UAVSAR Flight Dates | UAVSAR Flight Time (Local) | Field Survey Dates | GPR Survey Time (Local) | Ground Observations |
|---|---|---|---|---|
| 12 February 2020 | 11:10 | 12 February 2020 | 12:06 | GPR, SP, PD |
| 19 February 2020 | 11:42 | 19 February 2020 | 11:11 | GPR, SP, PD |
| 26 February 2020 | 11:24 | 26 February 2020 | 14:55 | GPR, TLS, SP, PD |
| 12 March 2020 | 10:54 | 11 March 2020 | 9:51 | GPR, TLS, SP, PD |
| 15 January 2021 | 10:43 | 15 January 2021 | 11:12 (MR), 14:49 (CP) | GPR, SP, PD |
| 20 January 2021 | 12:20 | 20 January 2021 | 11:18 (MR), 15:33 (CP) | GPR, SP, PD |
| 27 January 2021 | 11:52 (141°), 11:35 (321°) | 27 January 2021 | 11:27 (MR), 15:21 (CP) | GPR, SP, PD |
| 3 February 2021 | 10:51 (141°), 10:34 (321°) | 2 February 2021 | 10:52 (MR), 14:01 (CP) | GPR, SP, PD |
| No flight | - | 10 February 2021 | - | TLS, SP |
| 23 February 2021 | 15:50 | 24 February 2021 | 10:59 (MR), 14:34 (CP) | GPR, TLS, SP, PD |
| 3 March 2021 | 9:13 | 3 March 2021 | 11:05 (MR), 14:43 (CP) | GPR, SP, PD |
| 10 March 2021 | 8:46 | 9 March 2021 | 11:01 (MR), 13:29 (CP) | GPR, SP, PD |
| 16 March 2021 | 9:03 | 18 March 2021 | 10:14 (MR), 14:24 (CP) | GPR, SP, PD |
| 22 March 2021 | 8:43 | 22 March 2021 | 10:31 (MR), 14:12 (CP) | GPR, SP, PD |

The region has a continental snow climate (e.g., Trujillo and Molotch, 2014), with a prairie snowpack at lower elevation (<2800 m) within the North Park region and montane and alpine snowpacks in the higher elevation Medicine Bow Mountains and Never Summer Range. Four SNOTEL stations and three automated stations that measured snow depth were located within the flight line (Figure 1a). The Joe Wright SNOTEL station, which was within 1.5 km of our field sites, receives a median peak SWE of 632 mm that occurs on a median date of 5 May (1979–2023). Vegetation within the flight line primarily consists of evergreen forest (58%) and shrubs (32%; Buchhorn et al., 2020). Engleman spruce (*Picea engelmanii*), subalpine fir (*Abies lasiocarpa*), and lodgepole pine (*Pinus contorta*) are the primary constituents of the forest, with interspersed Aspen (*Populus tremuloides*) groves (Fassnacht et al., 2018). From August to November 2020, the Cameron Peak fire burned >80 km$^2$ of the flight line, including the Cameron Peak field site (CP; figure 1a) region (McGrath et al., 2023), which is not accounted for in these land cover estimates.

During SnowEx 2020, we surveyed the Michigan River field site (MR; Figure 1b), located in mostly open meadows vegetated by willows and grasses, though spruce/fir forests with <70% canopy cover inhabited portions of the northern and southern extent of the GPR transects. We measured stratigraphy, density, snow depth, and snow temperature in two snow pits (MR1, MR2; Figure 1b), following the SnowEx methodology outlined by Mason et al. (2023). Interval boards, which captured snow accumulation between surveys, were installed within 10 m of MR1 and at the nearby Joe Wright SNOTEL station. We recorded new snow depth, SWE, and density at each interval board on each site visit. Repeat GPR surveys (~1.6 km in length; McGrath et al., 2021) were performed using a Sensors & Software PulseEkko 1.0 GHz GPR coupled to the snow surface via a sled and pulled behind and to the side of a snowshoer. Snow depths were probed every ~3 m along the GPR transect. Two snow-on terrestrial lidar scans were performed on 26 February and 12 March 2020, in addition to a snow-off UAV-borne lidar scan performed in August 2020 (Williams, 2021).

For SnowEx 2021, we expanded our surveys to include the Cameron Peak field site (CP; Figure 1c). At MR, GPR surveys (0.8 km in length; Bonnell et al., 2022) were altered to form a loop around the primary meadow, with a co-located snow pit (MR1) and interval board. Snow pits and interval boards were surveyed following the SnowEx methodology. Snow depths were manually probed along the eastern portion of the GPR transect at ~5 m intervals. We expanded to CP to leverage the reduced vegetation due to the Cameron Peak fire. CP has severely burned spruce/fir forest to the north and east, with an unburned stand in the central to southern portion (Figure 1c). A single snow pit and interval board was surveyed near the GPR transect (1.6 km in length) in the burned section. Snow depths were probed every ~5 m along the southeastern GPR transect, with ~200 m in the forest and ~200 m in the burned area. An automated station was installed near the CP snow pit, which measured snow depth, wind speed and direction, radiation, temperature, and soil moisture. Two snow-on terrestrial lidar scans were performed at both field sites on 10 February and 24 February 2021, with a snow-off terrestrial lidar scan performed on 27 May 2021 at CP (Williams, 2021).

## 3 Methods

### 3.1 UAVSAR processing

Here, we provide an overview of the key UAVSAR processing steps. For additional and more detailed information, we direct readers to Appendix A.1–A.2. During the 2020 and 2021 airborne campaigns, UAVSAR deployed a fully polarized L-band (1.26 GHz center frequency, 0.24 m wavelength), 80 MHz bandwidth, left-looking InSAR. The instrument was flown at an altitude of ~12,500 m and acquired data along a ~40 km stretch with a 16 km swath width (area = ~640 km$^2$; Figure 1a; NASA UAVSAR, 2023). In 2020, overpasses were performed with a temporal baseline of seven days for the first three acquisitions (12, 19, & 26 February) and 15 days for the final acquisition (12 March). In 2021, overpasses had varying temporal baselines (typically five to eight days) and due to other aircraft commitments, one acquisition had a longer baseline (20 days for 3–23 February). Poor coherence prevented phase unwrapping at the field sites for one InSAR pair (10–16 March 2021). The UAVSAR team at the NASA Jet Propulsion Laboratory processed the UAVSAR data and generated geocoded

amplitude, interferogram, unwrapped interferogram, and coherence products at ~5 m spatial resolution. We accessed the products from the Alaska Satellite Facility (ASF; NASA UAVSAR, 2020, 2021) and converted the products to geotiffs using

uavsar_pytools (Hoppinen et al., 2022). InSAR measures phase deformation within a single $\pm\pi$ radians phase cycle, which equates to about $\pm108$ mm SWE. Interferograms where $\Delta$SWE exceeded a full phase cycle for some pixels require unwrapping for the accurate estimation of $\Delta$SWE. Therefore, we focused on the unwrapped interferogram and coherence products, and outline key workflow steps for calculating $\Delta$SWE, rather than total SWE, in Fig. 2. Although we included all four polarizations, we present the horizontal-transmit/horizontal-receive polarization (HH) for all intervals except the 3–23

February 2021 interval, which used the vertical-transmit/horizontal-receive polarization (VH) due to incomplete phase unwrapping in the HH data product. Detailed radar SWE retrieval methodology is outlined in Appendix A.2.

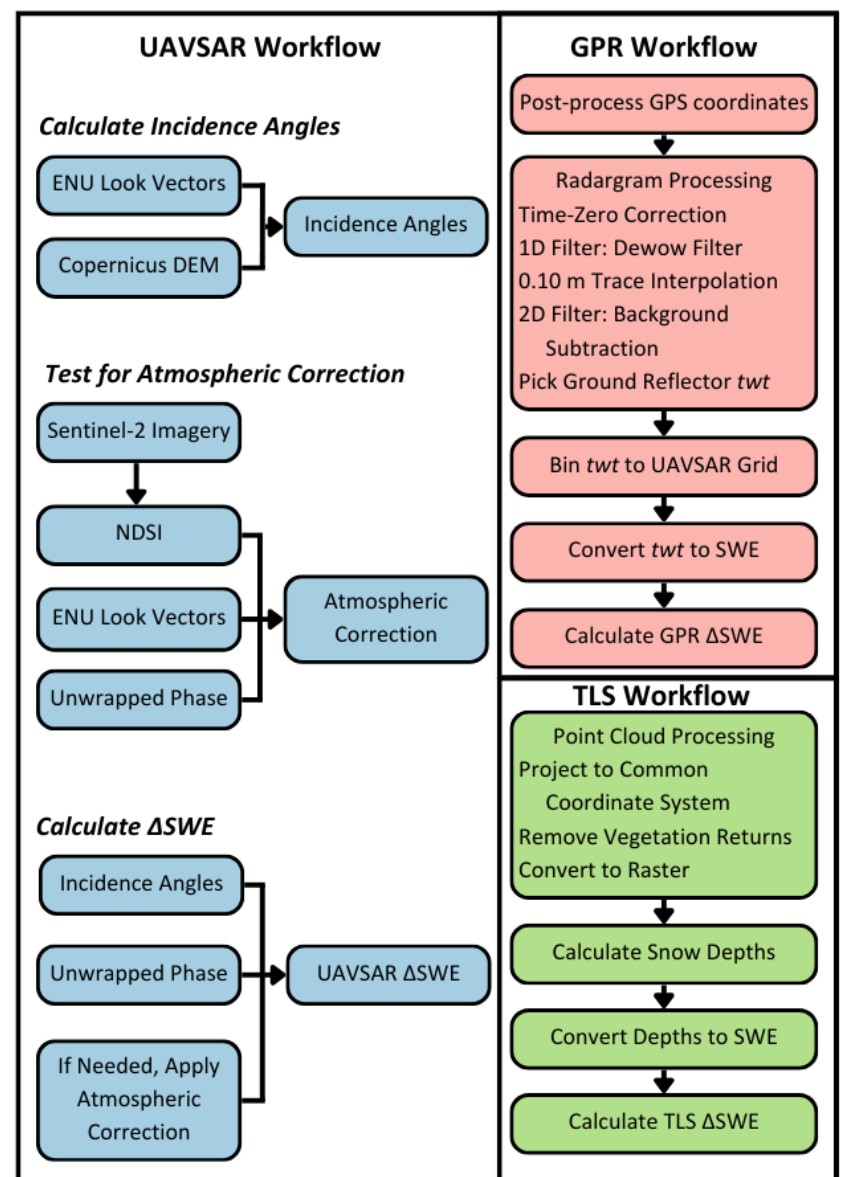

**Figure 2:** Workflow diagrams for deriving ΔSWE from UAVSAR, GPR, and TLS products. For simplification, UAVSAR workflow is described in three steps. ENU indicates the east, north, and up look vectors provided by UAVSAR.

We tested for atmospheric delays following methods developed by Tarricone et al. (2023). We identified snow-free pixels in the unwrapped interferograms using the normalized difference snow index (NDSI; Dozier, 1989) calculated from Sentinel-2 imagery (European Space Agency, 2022; Figure S1) and regressed snow-free unwrapped phase pixels against the corresponding signal path lengths. Importantly, this method assumes that snow-free pixels are not undergoing any physical changes that would lead to a phase change. We tested whether an atmospheric correction was needed using three criteria outlined in Appendix A.2.2. Importantly, no unwrapped interferograms met all three criteria (Table S1). Therefore, we conclude that stratified atmospheric artifacts are either limited for all interferometric pairs or were more complicated than what our linear model identified. See Appendix A.2.2 for a more detailed description of the atmospheric correction.

For these flights, UAVSAR had average look angles of 26–70° from near to far range. We calculated incidence angles in uavsar_pytools (Hoppinen et al., 2022; Equation A4) from the Copernicus 30 m DEM (rescaled to the UAVSAR grid) and the UAVSAR-provided look vector. The Copernicus DEM was chosen because it is the primary DEM used within the processing flow of ASF HyP3 and will be the basis for NISAR interferometric products. We evaluated incidence angles derived from the Copernicus DEM and the ΔSWE retrieval uncertainty caused by these incidence angles in Appendix A.2.5.

UAVSAR acquisitions were collected during the winter over relatively short temporal baselines (< 21 days). Therefore, we consider changes at the snowpack surface to be the primary driver of phase deformation in the unwrapped interferograms, but we provide a discussion of other potential sources of phase deformation in Appendix A.2.1. Changes at the snow surface may include new snow accumulation, sublimation, redistribution, or melt. For both study periods, we conclude that the snowpack is dry, based on results presented in Section 4.1. Thus, for ΔSWE retrievals, we consider only the density of snow that accumulated between UAVSAR acquisitions. Surface densities were estimated by averaging density measurements of the snow that accumulated on the interval boards between UAVSAR acquisitions (Section 3.2). For instances where snow accumulation had occurred but had been removed from the interval board by, for example, wind redistribution, we used an average of the uppermost 10 cm of the snow pit-measured densities. For each interferometric pair, we converted surface densities to relative permittivity (Equation A5). Relative permittivities, unwrapped phase, and incidence angles were then used to calculate snow depth changes (Equation A6), which were subsequently converted to ΔSWE using the surface snow density (Equation A7). Because InSAR phase is relative (Woodhouse, 2017), we estimated absolute phase as the median difference between a 20% set of randomly selected GPR ΔSWE retrievals (Section 3.2) and coincident UAVSAR ΔSWE retrievals for each interval. The median differences were then subtracted from the UAVSAR ΔSWE retrievals for each interval and the 20% of the GPR observations used in this step were removed from subsequent analyses. Finally, we supplemented our analysis by evaluating an InSAR ΔSWE retrieval method that approximates ΔSWE from the InSAR phase change and the incidence angle (Leinss et al., 2015) and is thus independent of snow density and relative permittivity measurements. The methods and results of this analysis are reviewed in Appendix A.2.4.

UAVSAR coherence values from corresponding TLS and GPR pixels were used to evaluate coherence as a measure of noise for ΔSWE retrievals. Coincident GPR and UAVSAR ΔSWE retrievals were binned by coherence and the root mean squared error (RMSE) of the UAVSAR ΔSWE retrievals was calculated for each bin. The effect of temporal baseline upon

coherence and UAVSAR ΔSWE retrieval accuracy was then evaluated by calculating the median coherence and RMSE for UAVSAR ΔSWE retrievals across all temporal baselines used in this analysis.

## 3.2 Processing ground-based measurements

### 3.2.1 In situ measurements

Key in situ measurements included snow pit temperatures, pit-measured densities, pit-measured depths, interval board densities and SWE, and manually probed depths. Pit-measured temperatures were used to detect the possible presence of liquid water within the snowpack. Pit-measured densities were averaged to estimate bulk density, which was used in SWE calculations for the snow pits, GPR, TLS, and probed depths. Interval board densities were used for ΔSWE calculations in the UAVSAR workflow, however, for some dates, the interval boards yielded little-to-no accumulation due to wind redistribution or a lack of precipitation. For these dates, the pit-measured densities from the upper 0.10 m of the snowpack were averaged and used in the UAVSAR workflow. Probed depths were not repeated in identical locations but were geocoded using a Geode GNS2 receiver mounted on top of the probe and converted to SWE using the bulk snow densities. Because the probed depths had a sampling of 1–2 measurements per UAVSAR pixel and were not collected in repeated locations, we used the depth probe dataset to evaluate the GPR and TLS SWE accuracy, rather than evaluating the UAVSAR ΔSWE retrievals directly.

### 3.2.2 GPR

GPR locations were collected via an Emlid RS2 GNSS receiver onboard the GPR sled and post-processed with an Emlid base station located at the MR field site to ensure a spatial accuracy of <0.25 m. High accuracy is important, given that these transects were repeated and the product of interest is ΔSWE, which is sensitive to geolocation errors. Radargrams were processed in ReflexW (Sandmeier, 2019) in four general steps: (1) apply time-varying time-zero correction, (2) one-dimensional de-wow filter to remove low-frequency noise, (3) trace interpolation to ~0.10 m, and (4) two-dimensional filter to remove instrument noise. After processing the radargrams, the ground reflector, identified as the highest magnitude positive amplitude reflector at depth, was picked and its corresponding two-way travel time (*twt*), representing the time-of-flight through the snowpack, was exported. Further GPR collection and radargram processing details are presented in McGrath et al. (2021) and Bonnell et al. (2022). Bulk snow density was then estimated as the average bulk density between available snow pits and used to estimate bulk relative permittivity (Equation A5) and, thereby, the velocity of the radar signal (Equation A9). Using the estimated velocity, we converted *twt* to SWE (Equations A10, A7). A detailed summary of the GPR theory and methods is provided in Appendix A.3. We evaluated the accuracy of GPR SWE retrievals through a comparison with SWE from probed depths by calculating the median GPR SWE retrieval within a 1.5 m radius around each probed depth. GPR SWE retrievals were then binned at the spatial resolution of the UAVSAR grid by taking the median

value of all points within each grid cell. SWE retrievals from corresponding dates were then differenced to generate GPR ΔSWE. The GPR workflow is summarized in Fig. 2.

### 3.2.3 Lidar Scans

Repeat snow-on terrestrial lidar scans were performed in 2020 on 26 February and 12 March at the MR site and in 2021 on 10 February and 24 February at the MR and CP sites. Snow-off lidar scans include a UAV-borne lidar scan that was performed for the MR site in August 2020 and a terrestrial lidar scan performed for the CP site on 27 May 2021. Terrestrial lidar scans were aligned and georeferenced by UNAVCO, Inc. (Williams, 2021). The USGS processed a bare-earth digital elevation model (DEM) from the UAV-borne lidar scan (Bauer et al., 2023). Lidar point clouds were reprojected and surface or bare ground returns were classified. These points were then converted to rasters, gridded and aligned to the UAVSAR grid, using the average elevation value per pixel. We derived snow depths for each snow-on scan date by subtracting snow-free rasters from snow-on rasters. Snow depth rasters were converted to SWE using the bulk density from the snow pits. ΔSWE was calculated for 26 February to 12 March 2020 and for 10–24 February 2021 by differencing the corresponding SWE rasters. To align TLS datasets with the 3–23 February 2021 InSAR pair, we subtracted the SWE measured on the interval board between 2–10 February 2021 from the UAVSAR ΔSWE retrievals. TLS ΔSWE was then directly compared with the UAVSAR ΔSWE retrievals. The terrestrial lidar workflow is summarized in Fig. 2.

### 3.3 Comparison between UAVSAR and automated stations

We obtained daily observations of snow depth, SWE, and air temperature from the Joe Wright SNOTEL station (ID 551) and daily observations of SWE from an additional three SNOTEL stations within the UAVSAR swath for the 2020 and 2021 seasons (Figure 1a; Table S2). Daily snow depths were obtained from three automated stations (two with sonic sensors and one with a snow stake paired with a time-lapse camera) within 4.5 km of the Joe Wright SNOTEL station (Table S2). We converted the snow depths to SWE by calculating density from Joe Wright SNOTEL station measurements of SWE and snow depth. SWE estimates were then smoothed with a five-day moving median filter to reduce the effects of new snow settlement.

We expanded our UAVSAR analysis beyond our relatively small field sites (~0.2 km$^2$ total area) to include measurements from the four SNOTEL stations and three automated stations within the swath (Table S2). We calculated the median UAVSAR ΔSWE within a 3×3 pixel grid (~15 m x ~15 m) around each station, added the ΔSWE retrievals for each interval, and matched the ΔSWE time series to the station time series by adding the station's SWE at the start of the UAVSAR flights to the UAVSAR ΔSWE time series. Because of spatially extensive missing data within the 10–16 March 2021 interferometric pair, we adjusted the UAVSAR ΔSWE time series at each station with the ΔSWE measured by the station. Median coherence was calculated within each 3×3 grid for the SNOTEL stations to evaluate the effects of coherence upon the ΔSWE retrieval time series. Last, station-measured SWE was compared with cumulative InSAR SWE for the final dates of the 2020 and 2021 UAVSAR acquisitions.

# 4 Results

## 4.1 Field observations of SWE and snow density

UAVSAR flights coincided with 117 mm of SWE accumulation (18% of peak SWE; Figure 3a) during the 2020 campaign (4 weeks) and 282 mm of accumulation (48% of peak SWE; Figure 3b) during the 2021 campaign (9 weeks). SWE at the in situ interval boards increased on average by 34 ±12 mm and 31 ±29 mm per flight interval during the 2020 and 2021 campaigns, indicating that ΔSWE at the field sites was likely within a full phase cycle (±108 mm; Appendix A.2.3) for most UAVSAR acquisitions. New snow density, used for UAVSAR ΔSWE calculations, ranged between 106 and 145 kg m$^{-3}$ across all study dates in 2020 (Figure 3c), and over a larger range, 118–219 kg m$^{-3}$, in 2021 (Figure 3d). Bulk density, used for GPR and TLS SWE calculations, increased minimally between most flights (mean = +20 kg m$^{-3}$; Figure 3c–d), with a notable exception being the 12–19 February 2020 pair (mean = +72 kg m$^{-3}$).

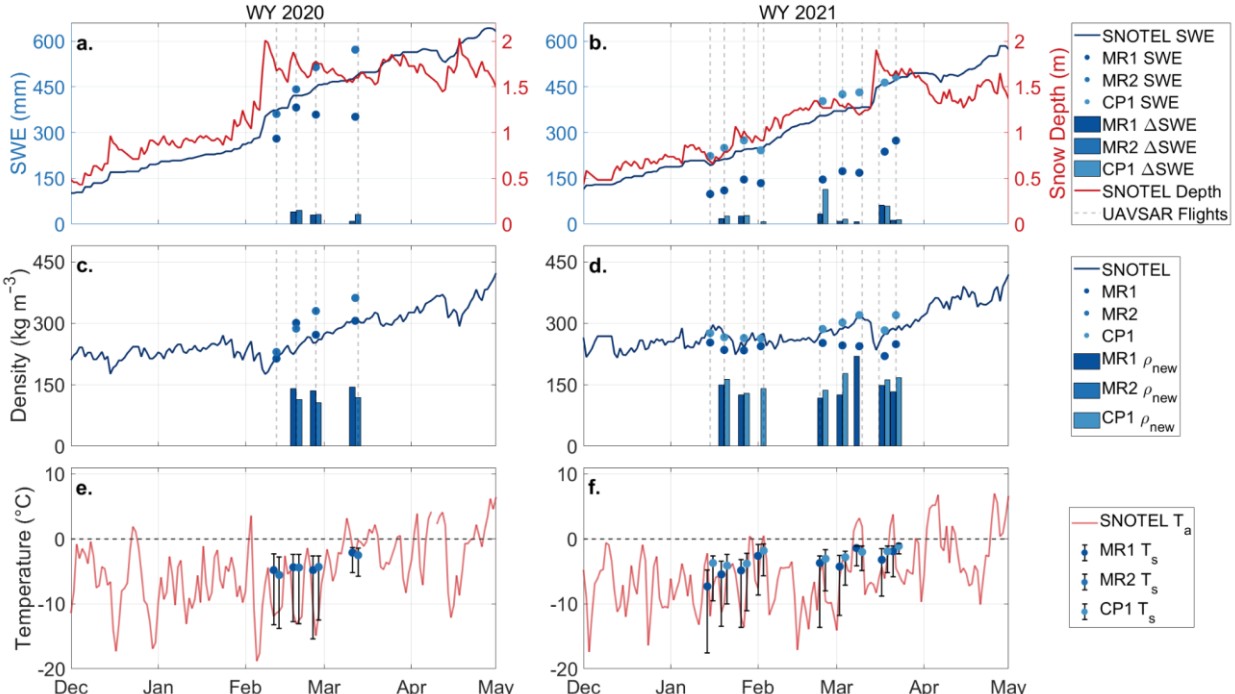

**Figure 3:** Joe Wright SNOTEL SWE and snow depth, bulk SWE and interval board SWE (ΔSWE) recorded at snow pits MR1, MR2, and CP1 for (a) water year (WY) 2020 and (b) WY 2021. SNOTEL density, bulk density and interval board density ($\rho_{new}$) recorded at snow pits for (c) WY 2020 and (d) WY 2021. SNOTEL air temperature ($T_a$) and error bar plots of snow pit temperatures for (e) WY 2020 and (f) WY 2021. UAVSAR acquisitions are represented as vertical dashed gray lines for plots (a–d). Bar graphs and error bar plots are paired and centered on the field survey date. Error bar plots show the median and the 25 and 75% quantiles.

Surface melting can lead to significant decorrelation of the radar signal and cause increased uncertainty in the ΔSWE retrievals. There were three notable warm periods during the campaigns (7–9 March 2020, 2–10 March 2021, and 21–22

March 2021), but median snow pit temperatures during our survey dates remained <−1.1°C (Figure 3e–f). We did observe near-surface melt-freeze crusts in the snow pits during certain surveys, but our observations suggest that liquid water content

was absent or minimal during UAVSAR flight times (Table 1) at our study sites throughout the campaigns.

GPR SWE retrievals from the 2020 MR field site showed that median SWE increased by 127 mm between 12 February and 11 March (Figure 4a), with the largest median ΔSWE occurring during the 12–19 February interval (+99 mm). The 2021 MR (Figure 4b) and CP (Figure 4c) field sites showed similar dynamic ranges, with GPR SWE retrievals increasing by 249 mm at the MR site and 233 mm at the CP site between 15 January and 22 March. For both sites, the largest median ΔSWE

occurred during the 2–24 February interval (MR = +97 mm, CP = +110 mm). GPR SWE retrievals and SWE converted from depth probe measurements are highly correlated, with an overall Pearson's correlation coefficient (r) of 0.97 and an overall RMSE of 35 mm (Figure S2).

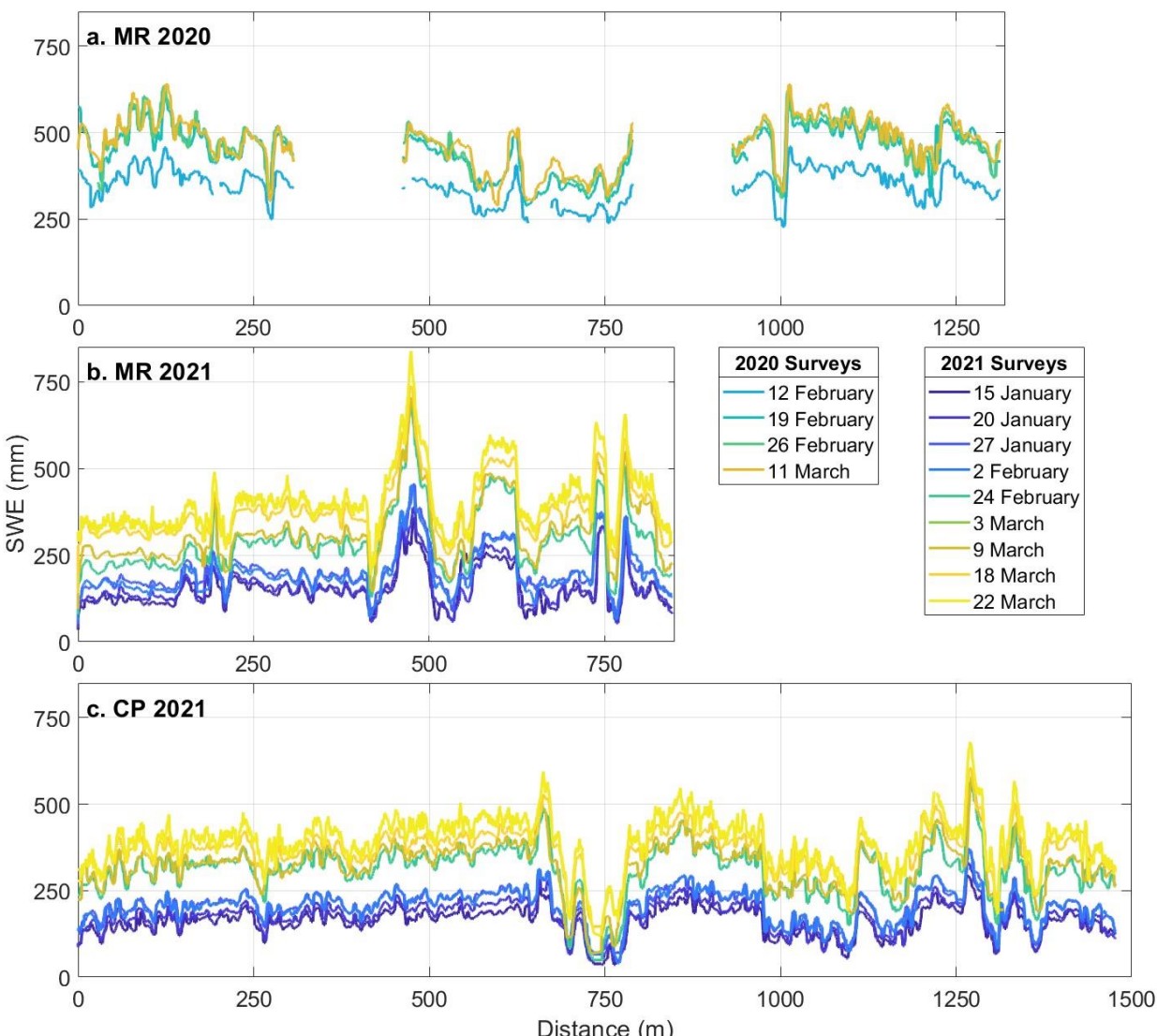

**Figure 4:** GPR SWE retrievals from the (a) 2020 and (b) 2021 MR field site, and (c) the 2021 CP field site. For (a), the transect begins at 0 m at the southern transect terminus and progresses northward (Figure 1b). For (b), the transect starts at 0 m at the northeast corner and progresses clockwise (Figure 1b). For (c), the transect starts at 0 m at the southeast corner and progresses counter-clockwise (Figure 1c). GPR SWE retrievals in (a–c) have been smoothed with a 5 m moving median filter.

**4.2 UAVSAR ΔSWE retrievals at the field sites**

UAVSAR ΔSWE retrievals along the GPR transect at the 2020 MR field site saw a mean cumulative increase of 40 mm for the three intervals (Figure 5c–e; Table S3). The largest median ΔSWE occurred during the 12–19 February interval (median

= +97 mm), with modest SWE increases observed for both the 19–26 February (median = +16 mm) and 26 February to 12 March (median = +8 mm) intervals. The largest ΔSWE retrieval range was observed for the 12–19 February interval (minimum = +60 mm, maximum = +149 mm). The expanded 2.7 km × 3.6 km region around the MR site reveals a somewhat different pattern than ΔSWE retrievals along the transect, with less accumulation for 12–19 February (+67 mm)

and negligible SWE changes for 19–26 February (0 mm) and 26 February to 12 March (+1 mm; Figure 5c–e).

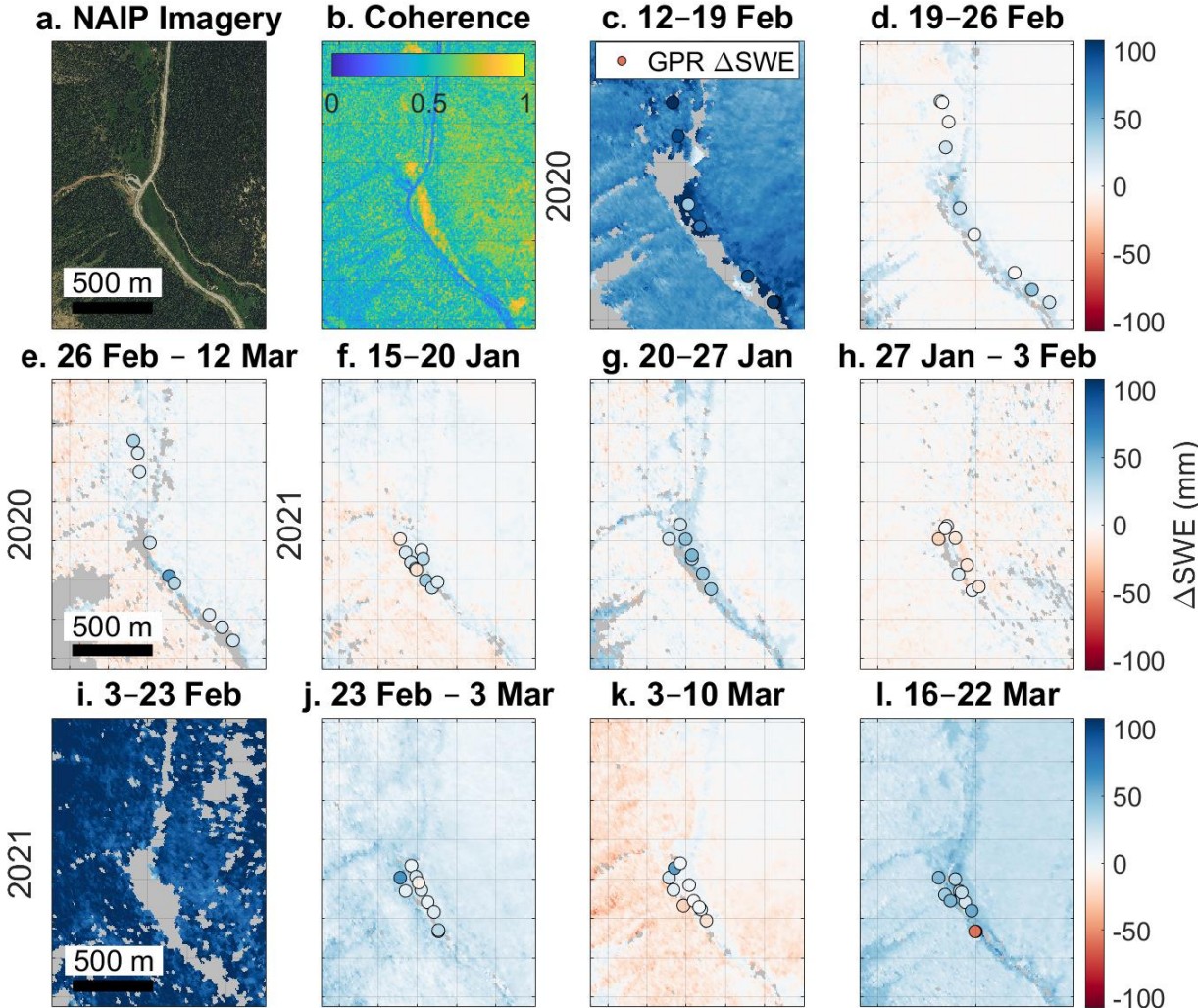

**Figure 5:** (a) National Agriculture Imagery Program (NAIP) imagery from Summer 2023. (b) Median coherence across all dates. (c–l) UAVSAR ΔSWE retrievals for each 2020 and 2021 date interval at the MR field site. GPR ΔSWE retrievals are overlain, but reduced to 5% of the total sample size for visual clarity. ΔSWE colors are minimized/maximized at approximately one phase cycle (±108 mm). All

355 dates used the 141° flight heading and HH polarization, except for the 27 January to 3 February 2021 interval which used the 321° heading and the 3–23 February 2021 interval which used the VH polarization. No GPR points are visible for the 3–23 February 2021 interval because no coincident InSAR ΔSWE retrievals were successfully unwrapped.

UAVSAR ΔSWE retrievals along the GPR transects at the 2021 MR field site saw a median cumulative increase of 104 mm for six of the seven 2021 intervals (no data for 3–23 February 2021), whereas the median cumulative increase for the expanded 2.7 km × 3.6 km region was +143 mm (Figure 5f–l). At the CP site, the median cumulative SWE across the seven surveys was 203 mm along the GPR transect and 171 mm from the 2.2 km x 3 km expanded region around the CP field site (Figure 6c–i). The largest median ΔSWE for the expanded regions occurred during the 3–23 February interval (MR median = +103 mm, CP median = +107 mm). Minimum UAVSAR ΔSWE retrieval medians from the expanded regions were observed on 27 January to 3 February at CP (median ΔSWE = −2 mm) and 3–10 March at MR (median ΔSWE = −6 mm).

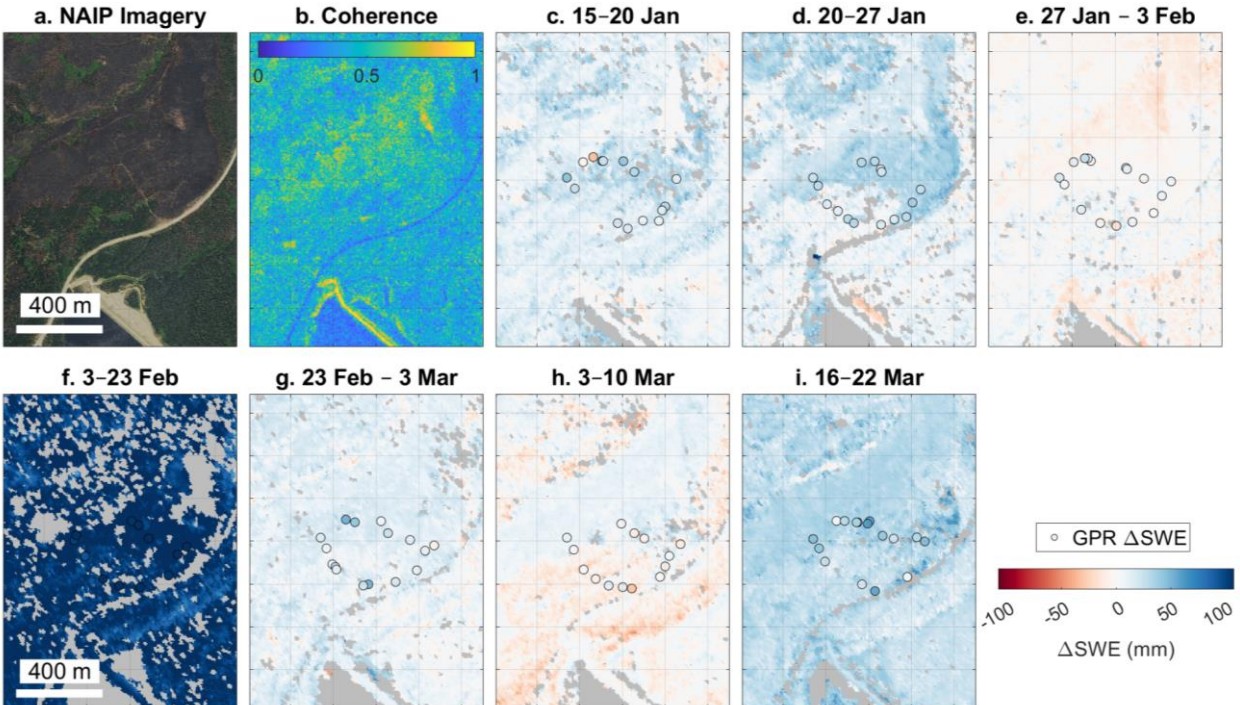

**Figure 6:** (a) Summer 2023 NAIP Imagery of the CP study site. (b) Median coherence across all dates. (c–i) UAVSAR ΔSWE retrievals for each 2021 date interval at the CP field site. GPR ΔSWE retrievals are overlain but reduced to 5% of the total sample size. ΔSWE colors are minimized/maximized at approximately one phase cycle (±108 mm). All dates used the 141° flight direction and HH polarization, except for the 27 January to 3 February interval which used the 321° direction and the 3–23 February interval which used the VH polarization.

UAVSAR ΔSWE retrievals appear to capture detailed spatial distributions of ΔSWE across all dates at each field site. In particular, larger SWE accumulation is observed in the open meadows and avalanche paths in the MR study area than the surrounding forests (mean difference = 66%, range of mean differences = −2 to +29 mm; Figure 5). These patterns are particularly noticeable at the MR site for the 12–19 February 2020 interval (Figure 5c), which recorded a median ΔSWE increase of +98 mm in open meadows and avalanche paths, whereas ΔSWE in the surrounding forests increased by a median of +69 mm. A similar spatial pattern exists at the CP site, as the burned area consistently recorded a larger ΔSWE than

adjacent unburned forests. This is best observed in the 20–27 January 2021 interval (Figure 6d). During this interval, we calculated an average of 31 mm ΔSWE in the burned area and 15 mm ΔSWE in the unburned forests. Median coherence across the time series is somewhat higher for unforested areas in both the MR and CP field sites (+0.05; Figures 5b,6b). This subtle difference is further illustrated within the CP field site, where median coherence of the seven-day baseline InSAR pairs increased from 0.56 in 2020 pre-burn forests to 0.60 in 2021 post-burn areas (p = <0.0001).

### 4.3 Evaluating UAVSAR ΔSWE retrievals with GPR

UAVSAR ΔSWE retrievals have a relatively low pixel-wise correlation with GPR ΔSWE retrievals for any single InSAR pair (r = –0.24 to 0.20; Table S4). However, compiling the measurements across all surveys increases the ΔSWE dynamic range and correlation substantially (r = 0.79; Figure 7a). Here, we present a time series that includes only InSAR pairs from the HH polarization for all dates except the 3–23 February 2021 pair, which is represented by the VH polarization. For this time series, we observe RMSEs from 16–34 mm (Table S4) for single InSAR pairs, with an overall RMSE = 22 mm (Figure 7a). Although pixel-wise comparisons between UAVSAR and GPR ΔSWE retrievals exhibit scatter, the box plot distributions for ΔSWE at co-located GPR-UAVSAR pixels are nearly identical, yielding absolute median differences between median GPR ΔSWE and median UAVSAR ΔSWE of 0–4 mm (Figure 7b,c; Table S2). Although we primarily present analyses based on the HH polarization, we find that UAVSAR ΔSWE retrievals have nearly equivalent RMSE values across all four polarizations (RMSE = 19–22 mm; Table S4).

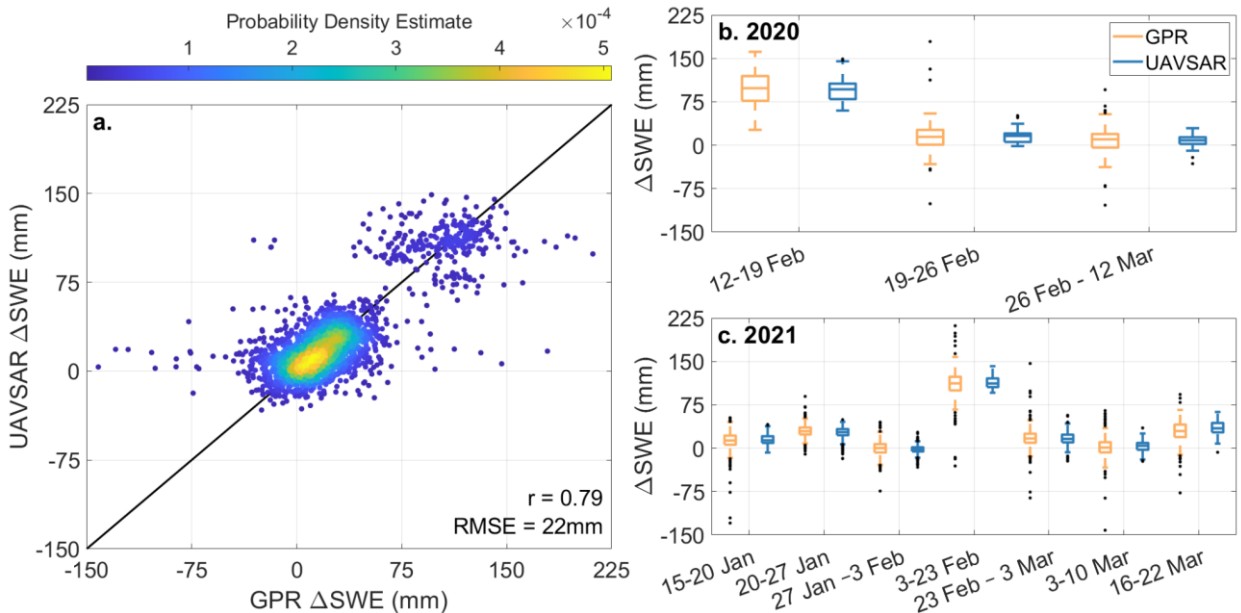

**Figure 7:** (a) UAVSAR ΔSWE retrievals compared with coincident GPR ΔSWE retrievals, with reported Pearson's correlation coefficient (r) and RMSE (n = 2833). Points in (a) are colored by point density. (b) 2020 box plot distributions of GPR and UAVSAR ΔSWE

retrievals paired by date. (c) 2021 box plot distributions of GPR and UAVSAR ΔSWE retrievals paired by date. Box plots show the median, 25th and 75th quantiles, and the maximum and minimum, with outliers (>1.5 times the interquartile range) shown as points.

We explored the possibility of coherence as an error metric for UAVSAR ΔSWE retrievals and found that RMSE exhibited a narrow range (21–25 mm) for coherence bins between 0.1 and 0.7 (Figure 8a). However, RMSE at very low coherence (0–0.1) is double the RMSE at very high coherence (0.9–1.0). Average coherence was highest for ~weekly baselines, but average coherence for the 15-day baseline (0.51) was within the range of average coherence for the five-to-eight-day temporal baselines (Figure 8b). Of note, the 20-day baseline had average coherence >0.40 (Figure 7d) but yielded the highest RMSE (33 mm; Figure 8b).

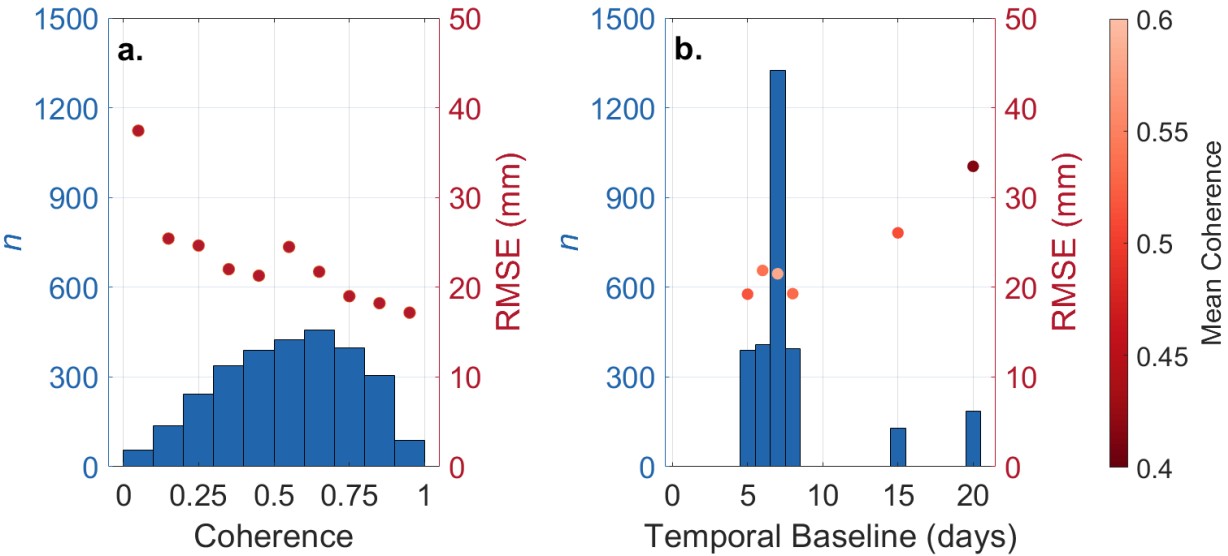

**Figure 8:** Histograms of (a) UAVSAR coherence values and (b) temporal baseline from co-located GPR and UAVSAR pixels. RMSE is shown for each bin. In (b), RMSE points are colored by mean coherence per temporal baseline bin.

## 4.4 Evaluating UAVSAR ΔSWE retrievals with TLS

TLS ΔSWE retrievals had median values of +9 mm for the MR field site during the 26 February to 12 March 2020 interval, and +55 and +39 mm at the MR and CP field sites during the 10–24 February 2021 surveys (Figure 9a,d,g). TLS SWE retrievals have a high correlation with SWE converted from depth probes, with a r of 0.83 and RMSE of 66 mm (n = 189; Figure S3). For each set of TLS acquisitions, UAVSAR ΔSWE retrievals had median values of +6, +60, and +45 mm, respectively (Figure 9b,e,h). Spatial patterns were similar between the two methods of ΔSWE retrievals. Large portions of data are missing in Fig. 9e due to coherence-related phase unwrapping errors. RMSEs were comparable between the 2020 survey (MR = 20 mm) and the 2021 surveys (MR = 15 mm, CP = 20 mm). UAVSAR ΔSWE retrievals have an overall

RMSE of 19 mm and an r of 0.72 when compared with TLS. Coherence was used to color points on the UAVSAR-TLS
comparison plots (Figure 9c,f,i) and shows that scatter is approximately equal throughout the range of observed coherences.

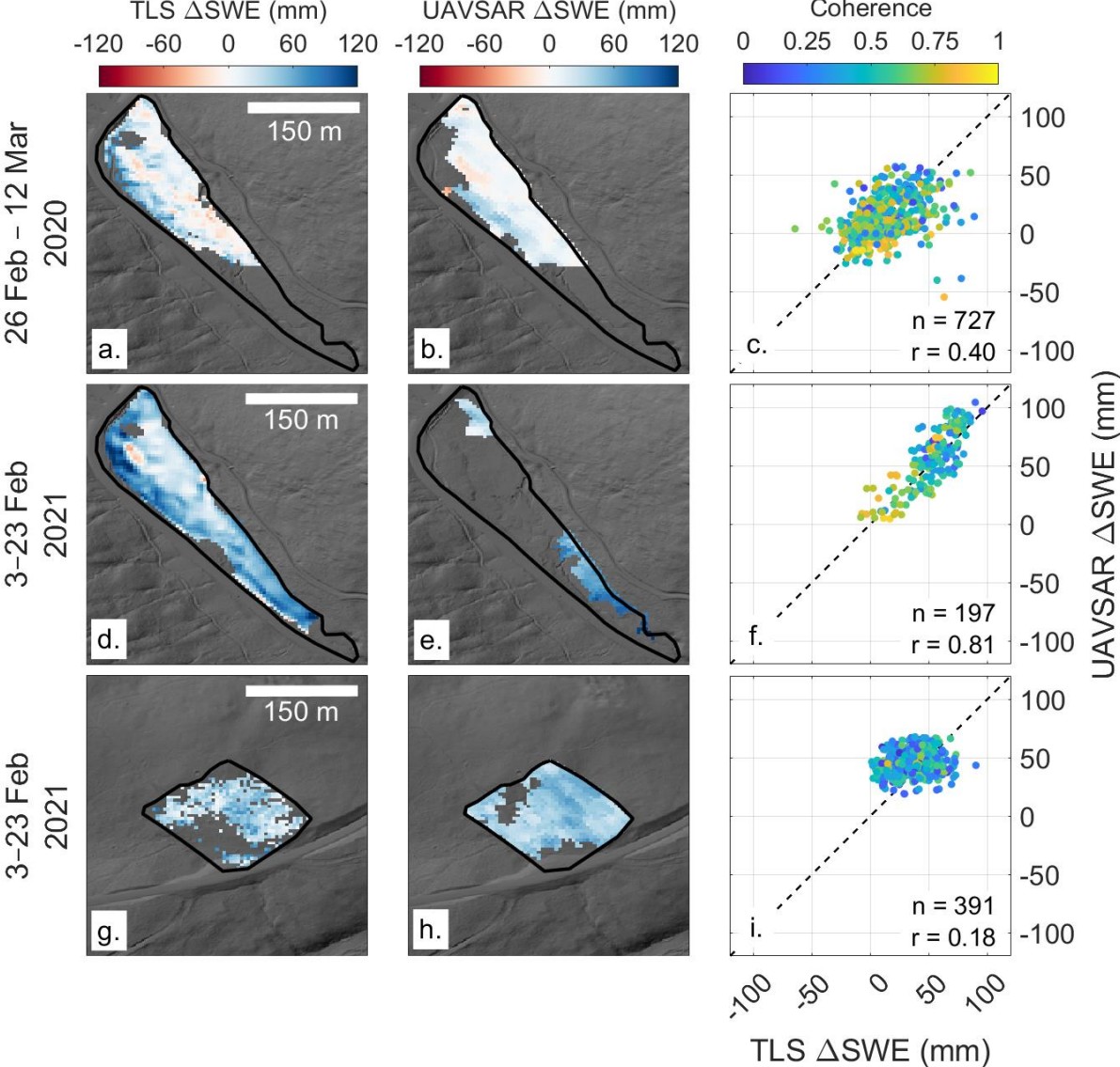

**Figure 9:** Results of the ΔSWE comparison between TLS and UAVSAR. Rows are organized by date and field site. Columns include TLS ΔSWE (left column), UAVSAR ΔSWE (middle column), and the comparison between TLS and UAVSAR (right column). SWE measured
at the interval board on 10 February 2021 was subtracted from UAVSAR ΔSWE for 3–23 February 2021 to align with the TLS survey dates. Comparison plots are colored by coherence. The number of pixels (n) and Pearson's correlation coefficient (r) are reported for each comparison.

## 4.5 Evaluation of UAVSAR time series at automated stations

UAVSAR SWE retrievals overestimated SWE accumulation for the 12–19 February 2020 InSAR pair by an average of 163% at the automated stations but underestimated SWE accumulation by an average of 88% between 19 February and 12 March (Figure 10a–e). 2021 cumulative UAVSAR SWE retrievals record net increases at all seven sites (+109 to +219 mm), which is similar to the net increases recorded by the stations (+101 to +242 mm; Figure 10a–g). Median coherence for the 2020 season was lowest at the Lake Irene SNOTEL station (median coherence = 0.30) and highest at the Phantom Valley SNOTEL station (median coherence = 0.63), whereas median coherence for the 2021 season was lowest at the Montgomery Snow Stake (median coherence = 0.49) and highest at the Lake Irene SNOTEL station (median coherence = 0.60). The lowest median coherence for all sites was observed for the 26 February to 12 March 2020 interval (median coherence = 0.31), an interval that yielded negative SWE retrievals for three of the five operating stations (–17 to –3 mm). At the end of the UAVSAR campaigns, cumulative UAVSAR SWE retrievals from the seven stations (n = 12) yielded an RMSE = 42 mm and an r = 0.92 (Figure 10h).

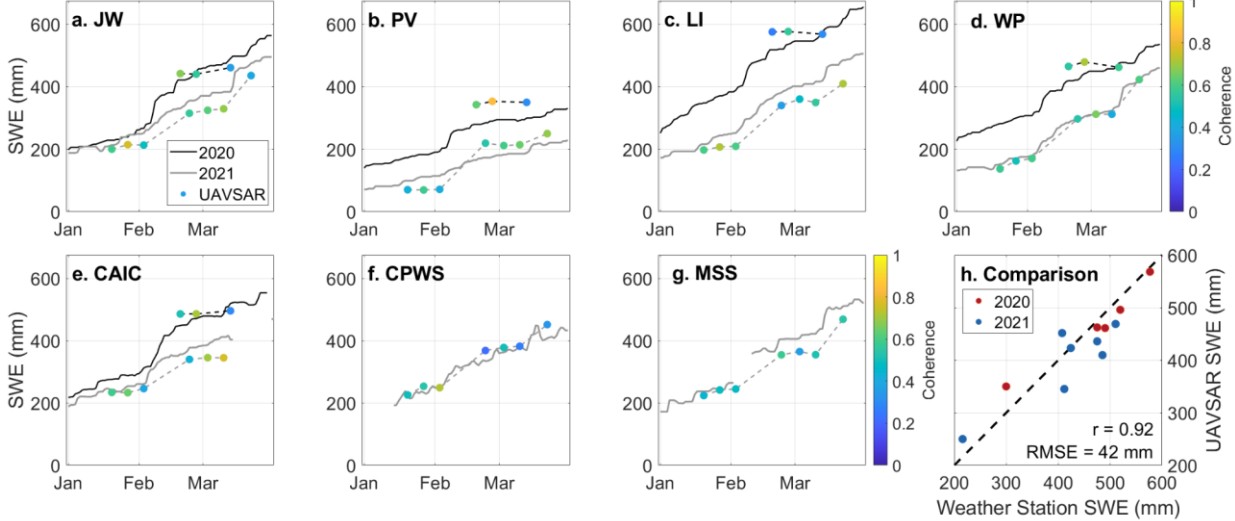

**Figure 10:** Time series of 2020 and 2021 UAVSAR ΔSWE retrievals compared with SWE from (a) the Joe Wright SNOTEL station (JW), (b) the Phantom Valley SNOTEL station (PV), (c) the Lake Irene SNOTEL station (LI), (d) the Willow Park SNOTEL station (WP), (e) the Colorado Avalanche Information Center weather station (CAIC), (f) the Cameron Peak field site weather station (CPWS), and (g) the Montgomery Snow Stake site (MSS). Mean 9-pixel coherence is shown for each UAVSAR point. (h) Comparison between SNOTEL SWE and cumulative UAVSAR SWE for the last UAVSAR flight for each year. For sites (e–g), only snow depth was observed and SWE was estimated using density recorded at the closest SNOTEL station (Joe Wright SNOTEL). Methods describing the alignment of the UAVSAR time series to the automated stations are described in Section 3.3.

# 5 Discussion

## 5.1 Accuracy of L-band InSAR ΔSWE retrievals

From our evaluation with GPR and TLS, we established the RMSE for L-band InSAR ΔSWE retrievals as 19–22 mm for single InSAR pairs (Figures 7,9). For cumulative InSAR SWE, we estimated a RMSE of 42 mm at seven automated stations (Figure 10). Previous studies have established that UAVSAR ΔSWE retrievals resemble the spatial patterns of lidar-derived ΔSWE retrievals, but differences between the two datasets were not systematic (Marshall et al., 2021; Palomaki and Sproles, 2023). Marshall et al. (2021) evaluated UAVSAR ΔSWE retrievals over a 4 km$^2$ relatively flat and non-forested region of

Grand Mesa, Colorado using airborne lidar and found very low error for the technique (RMSE = 9 mm). UAVSAR ΔSWE retrievals have been evaluated using GPR and automated station measurements in Valles Caldera, New Mexico (Tarricone et al., 2023) and from in situ and SNOTEL measurements of ΔSWE in the mountains of Idaho (Hoppinen et al., 2024). Both studies identified and corrected significant atmospheric artifacts and contained at least one InSAR pair that was collected when liquid water content was present in the snowpack, but estimated study-wide errors of similar magnitude found by our

study (RMSE = 15–40 mm; Table S4).

UAVSAR ΔSWE retrievals had higher RMSE in 2020 than in 2021 (Table S4), and the agreement between the InSAR time series and the automated stations was poorer in 2020 than in 2021 (Figure 10). One potential explanation for the lower agreement in 2020 was the significant deviation (>10 m) from the cross track and vertical baselines of the aircraft during the 2020 flights (Jones et al., 2016; NASA UAVSAR, 2023). UAVSAR ΔSWE spatial patterns are similar to those of TLS

ΔSWE (Figure 9) and the comparison of UAVSAR and GPR ΔSWE site-wide distributions reveal nearly identical medians (absolute median difference = 2 mm; Figure 7b–c). We found that low coherence did not substantially increase the RMSE of UAVSAR ΔSWE retrievals as the RMSE was less than 35 mm for >10-day temporal baselines (Figure 8). However, lower coherence for InSAR pairs with >10-day temporal baselines exhibited issues with phase unwrapping. Collectively, these findings suggest a high degree of accuracy and reliability for InSAR ΔSWE retrievals, particularly in relatively simpler

environments (i.e., dry snow conditions, non-forested areas, slopes <20°) and when atmospheric delays are limited.

## 5.2 Considerations for future evaluations of InSAR ΔSWE retrievals

The NISAR satellite mission holds promise for global repeat 12-day ΔSWE retrievals, providing the opportunity to evaluate the L-band InSAR technique in a range of environments and to better assess its uncertainties. In our evaluation, we used two

ground-based methods that many snow community researchers have access to and showed that both methods are capable of assessing InSAR ΔSWE retrieval accuracy. Both methods can be used to derive spatially continuous SWE measurements over large areas and are therefore advantageous over standard in situ SWE measurement methods (Holbrook et al., 2016; McGrath et al., 2019). Below we outline advantages, considerations, and challenges of GPR and TLS for InSAR evaluation.

Few methods match the sophistication of InSAR for change detection. Of the two techniques we employed in our evaluation, lidar is the most applicable for change detection (Deems et al., 2013), but its methodology for ΔSWE retrievals is not straightforward. There are two conceptual paths for ΔSWE retrievals from lidar: (1) subtraction of two repeat snow-on lidar elevation surveys or (2) subtraction of two bulk SWE datasets derived from lidar. The first option is complicated by snow compaction, while the second option requires accurate bulk snow densities and a snow-off bare-earth digital terrain model, which may be difficult to acquire in densely vegetated areas. We chose the second option because bulk density variability is less of a concern for the relatively small areas surveyed by the TLS (Bonnell et al., 2023). We found the best agreement between UAVSAR and TLS ΔSWE retrievals for surveys that were aligned on the same date, as differential SWE accumulation/redistribution increased uncertainty (Figure 9). Note that if the TLS platform is set up on top of the snowpack, accurate TLS ΔSWE retrievals may be hindered by small shifts in the TLS platform as it settles in the snow (Currier et al., 2019).

Repeat GPR transects also have several challenges. Our survey methodology involved marking our transects and post-processing the onboard GPS sensor (±0.25 m accuracy), but it is likely that our tracks were offset by ±1–2 m from the transect for some surveys. Further, as SWE increases throughout a season, the *twt* to the ground reflector increases, effectively increasing both the GPR horizontal footprint and the potential for clutter in the radargram (Daniels, 2004). Surface-coupled GPR has the potential to both compact the snow below the sled and remove snow from the surface (e.g., McGrath et al., 2019), which may further increase the uncertainty of GPR ΔSWE retrievals, particularly in low density snow on the surface of the snowpack. These complications may explain the low Pearson's correlation coefficients observed between UAVSAR and GPR ΔSWE retrievals for single InSAR pairs (r = –0.24 to +0.2; Table S4), as well as the low GPR-UAVSAR ΔSWE retrieval relation ($r^2$ <0.1) described by Tarricone et al. (2023). However, as our analysis shows, repeat GPR transects are effective at evaluating the InSAR technique if there is enough data collected across a range of SWE accumulation magnitudes (Figure 7).

A major difference between UAVSAR and planned NISAR interferograms is the spatial resolution (~5 m vs. 80 m), which may complicate future NISAR ΔSWE retrieval ground-based evaluations. GPR surveyed along transects scaled well to the resolution of UAVSAR, but a different survey design (i.e., spiral or grid) may be required to provide sufficient coverage of the NISAR pixels. Thus, GPR may have increased uncertainty in its scalability due to a difficulty of repeating complicated survey designs. Lidar is scalable to coarser resolutions (e.g., 50 m; Painter et al., 2016) and TLS and drone-mounted lidar (e.g., Feng et al., 2023) may be valuable tools for evaluating InSAR ΔSWE retrievals at small field sites. However, at larger scales, comprehensive airborne lidar surveys will be required to fully evaluate NISAR ΔSWE retrievals.

## 5.3 Remaining questions for the L-band InSAR ΔSWE retrieval technique

L-band InSAR has been seen as a promising technique for high resolution snow monitoring for over a decade (Deeb et al., 2011), yet insufficient opportunities existed for robust evaluations. In the last few years, airborne InSAR campaigns over seasonal snowpacks have created opportunities for a more thorough evaluation of this technique. Our study, and others, show

that this technique can have high accuracy, but there are several areas of uncertainty that need to be considered, including forested environments, wet snowpacks, complex topography that results in steep incidence angles, spatially varying atmospheric delays, and the integration of InSAR ΔSWE retrievals with other remote sensing methods and models.

Recent UAVSAR studies (Hoppinen et al., 2024; Marshall et al., 2021; Palomaki and Sproles, 2023; Tarricone et al., 2023), including this study, have largely focused on ΔSWE retrievals in open environments. We found that ΔSWE retrievals were 66% higher on average in the open areas around the MR field site than below forest cover. Forest canopy interception and sublimation may play a role in this signal, because this process is known to drive a 20–30% reduction of total snowfall at the nearby Fraser Experimental Forest (Montessi et al., 2004). On the other hand, a contrast between lower snow surface

densities in the forest compared with the potentially higher densities we measured in the open could explain a similar magnitude of the signal. Unfortunately, we are unable to validate the forest ΔSWE retrievals as only 20% of GPR observations in 2020 and 10% of GPR observations in 2021 were collected below spruce/fir canopy (15–70% canopy cover). Forests interfere with the radar signal, reducing coherence and potentially biasing retrievals, particularly for longer temporal baselines (Li et al., 2022; Ruiz et al., 2022). However, coherence only improved by +0.05 from forests to open areas at our

field sites and even the removal of canopy due to the Cameron Peak wildfire only increased coherence by +0.04. Thus, because of its canopy penetrative capabilities, the L-band InSAR ΔSWE retrieval technique may be the first satellite-based technique viable for SWE monitoring in forests.

At our site, UAVSAR flights occurred during the accumulation season when the snowpack was likely dry (Figure 3e–f). However, SWE monitoring is needed for snowpacks that accumulate at or near 0°C and for the melt season, making ΔSWE

retrieval evaluation prioritized in wet snowpacks. Liquid water in the snowpack raises both the real and imaginary components of relative permittivity, which decreases the snowpack radar velocity and increases absorption of the radar signal, causing decreased signal penetration (Tsai et al., 2019). Even if the backscattering interface is unchanged, reduced radar velocity causes ΔSWE retrieval overestimation if the liquid water content is not considered (Bonnell et al., 2021; Tarricone et al., 2023). Tarricone et al. (2023) evaluated ΔSWE retrievals with the Landsat fractional snow-covered area

product and found reasonable snowpack ablation over a 14-day period in Valles Caldera, New Mexico, but Hoppinen et al. (2024) found reduced ΔSWE retrieval accuracy in wet snowpacks. Wet snow detection techniques have been developed and implemented at C-band (e.g., Gagliano et al., 2023; Nagler and Rott, 2000; Nagler et al., 2016) and similar techniques should be evaluated at L-band frequencies (e.g., Park et al., 2014).

**6 Conclusion**

During the winters of 2020 and 2021, UAVSAR collected L-band InSAR datasets over 12 mountainous regions of the western United States, including continental snowpacks of Colorado, intermountain snowpacks of Idaho and Montana, maritime snowpacks of California, and shallow mountain snowpacks in New Mexico. At the Cameron Pass field site, we used extensive GPR and TLS to evaluate UAVSAR ΔSWE retrievals over a three-pair time series (4 weeks) that saw 121

mm SWE accumulation in 2020 and a seven-pair time series (9 weeks) that saw 206 mm SWE accumulation in 2021. Our analysis was not complicated by the presence of liquid water within the snowpack and we emphasized GPR and TLS collection in open areas at our field sites. Our results indicate accurate statistical distributions for the L-band InSAR method for areas without forest cover (absolute median difference = 2 mm compared to GPR), but low correlation coefficients (r = −0.24 to +0.2) for individual InSAR pairs warrants caution for ΔSWE interpretation at the single pixel scale. UAVSAR ΔSWE retrievals exhibited distinct and repeated spatial patterns relating to the land cover, as forests averaged 66% less ΔSWE per InSAR pair than open meadows, burned forests, and avalanche alleyways in forests. We expanded our in situ SWE observations to include seven automated weather stations distributed throughout the swath and highlighted the utility of the InSAR method for measuring cumulative SWE (RMSE = 42 mm), a requirement for any SWE remote sensing method. We found that the range in RMSE from coherences of 0.10–0.90 was <10 mm, indicating that low coherence does not necessarily inhibit the accurate retrieval of ΔSWE. Although our ground observations did not target forested areas, we found the median coherence in the forests averaged 0.05 less than in the open, suggesting ΔSWE retrievals may be viable in these environments but the location of the amplitude-center as forest cover increases remains an active question. Collectively, our study supports the use of L-band InSAR for measuring SWE in mountain snowpacks, further highlighting the potential for NISAR and other upcoming L-band SAR satellites to contribute substantially to global SWE monitoring.

## Appendix A: Radar for SWE retrievals

### A.1 L-band transmissibility

At L-band frequencies (1–2 GHz, ~0.25 m wavelength), dry snow is fully transmissible because of limited interactions between snow grains and the radar signal (Tsai et al., 2019). The bulk of reflected energy is returned from the snow-ground interface for areas without dense vegetation (Nagler et al., 2022), but uncertainty regarding the source of the primary backscattering surface increases with increased vegetation density because the L-band signal interacts with tree trunks, large branches, and dense vegetation (Ottinger and Kuenzer, 2020).

### A.2 The InSAR technique for ΔSWE retrievals

### A.2.1 Introduction to the InSAR technique

SARs emit polarized radar signals at a given frequency and narrow bandwidth and record the amplitude and phase of backscattered signal (Woodhouse, 2017). InSAR is a change-detection technique that calculates the phase change between two radar signals operating at identical wavelengths and polarizations. Guneriussen et al. (2001) proposed a method for removing the snow accumulation signal from interferometric pairs where at least one of the acquisitions occurred during the snow season. Their proposed method forms the basis for most published InSAR ΔSWE retrieval techniques and is the one we implement. We applied this technique to repeat airborne acquisitions and assume the phase deformation is primarily due

to the accumulation or redistribution of snow. We accessed unwrapped interferograms from the ASF Distributed Active Archive Center (DAAC). Interferograms were unwrapped by the UAVSAR team, following the Integrated and Correlation Unwrapping method (Goldstein and Werner, 1998). This technique relies on the interferometric coherence ($\gamma$), which is calculated as

$$\gamma = \frac{E[u_1 u_2^*]}{\sqrt{E[|u_1|^2]}\sqrt{E[|u_2|^2]}} \tag{A1},$$

where $E$ is the expected value of a given variable and $u_1$ and $u_2$ are the amplitudes for the two image pairs (Woodhouse, 2017).

In the case of snow, the amplitude center is assumed to be the snow-ground interface and any deformation in phase is expressed as

$$\varphi = \varphi_{flat} + \varphi_{topo} + \varphi_{atm} + \varphi_s + \varphi_{err} \tag{A2},$$

where the total interferometric phase change ($\varphi$) is expressed as the sum of the phase changes that arise from changes in the relative distance between the radar platform and the ground target for flat Earth ($\varphi_{flat}$) and topography ($\varphi_{topo}$), changes in the atmospheric conditions that cause signal delays ($\varphi_{atm}$), the changes in phase caused by the change in snow depth or SWE ($\varphi_s$), and phase changes caused by instrument noise ($\varphi_{err}$; Deeb et al., 2011). Instrument noise can manifest as random error or systematic error, which can result from a non-constant flight track (Jones et al., 2016). Topographic corrections are minimized by the UAVSAR instrument, as it performs acquisitions within a repeated 10 m tube, but both the topographic and flat Earth contributions towards total phase change are accounted for in the UAVSAR unwrapped interferograms. However, atmospheric delays, caused by changes in atmospheric pressure and water vapor mass that occur between acquisitions, may influence the interferometric phase change (Bevis et al., 1992).

**A.2.2 Atmospheric correction for UAVSAR**

Atmospheric delays are generally described as stratified or turbulent, where stratified delays are manifested as phase ramps or are correlated with topography and occur due to relatively homogeneous differences in atmospheric conditions, whereas turbulent delays are more difficult to identify and are caused by heterogeneous differences in atmospheric conditions (Hu and Mallorquí, 2019). Modeling atmospheric delays from airborne platforms is complicated, primarily due to the relatively coarse vertical resolution of most atmospheric reanalysis/forecast products that extends higher than the UAVSAR flight altitude (~12.5 km). Three recently developed methods may be applicable for our study: (1) a statistical approach that models delays assuming a stratified atmosphere (Tarricone et al., 2023), (2) an approach that integrates phase delays along the signal path using ERA5 atmospheric data (Hoppinen et al., 2024), and (3) modeling the turbulent delay from atmospheric pressure and precipitable water using the High Resolution Rapid Refresh Model (HRRR; Gong et al., 2013). We chose the Tarricone et al. (2023) approach, which has higher spatial resolution than either the ERA5 or HRRR methods, and developed a workflow to evaluate the need for a stratified atmospheric correction.

The workflow estimates an atmospheric correction as a best-fit plane across the UAVSAR scene, by regressing the unwrapped phase at snow-free pixels with the radar signal path length. Before the analysis, we defined requirements that the atmospheric correction had to meet in order to be implemented: (1) regression slope estimators needed to be identical across all four polarizations and the estimator's p-value needed to be <0.05, (2) coefficients of determination ($r^2$) were required to be >0.20, and (3) the root mean squared error (RMSE) of atmospherically corrected ΔSWE had to improve the RMSE of uncorrected ΔSWE by >20%.

Sentinel-2 Level 2A (Surface Reflectance) 2020 and 2021 products were accessed in Google Earth Engine at 10 m resolution. Clouds were removed for each image and an average image was composited for all Sentinel-2 acquisitions between UAVSAR flights. Normalized difference snow index (NDSI; Dozier, 1989) between green and shortwave infrared (SWIR) bands was calculated as

$$NDSI = \frac{Band_{green} - Band_{SWIR}}{Band_{green} + Band_{SWIR}}$$ (A3).

We then masked out forests from the scene using the Copernicus Global Land Cover 100 m dataset (Figure S1). Snow-free pixels were identified as NDSI < 0.2, based on visual inspection of the optical imagery. We then regressed the unwrapped phase at snow-free pixels against the radar signal path length to estimate a phase ramp for each InSAR pair. We calculated RMSE for both atmospherically corrected and uncorrected datasets using SNOTEL ΔSWE calculated from the four SNOTEL stations (Table S2) where we took the median of the nearest nine UAVSAR ΔSWE pixels but removed stations that had coherence <0.5. No single interferogram met our listed requirements (Table S1). We conclude that stratified atmospheric delays may be present, but do not substantially affect the accuracy of ΔSWE retrievals.

### A.2.3 Calculating InSAR ΔSWE retrievals

Assuming all other phase terms are accounted for (Equation A2), ΔSWE can be calculated from the snow phase term, the radar wavelength ($\lambda$; ~0.238 m), the local incidence angle ($\theta_{inc}$), and the relative permittivity ($\varepsilon_s$). Because the radar signal intersects the snowpack obliquely, the unwrapped phase must be projected to the surface normal using the local incidence angle. We calculated incidence angles in uavsar_pytools (Hoppinen et al., 2022) as

$$\theta_{inc} = (-\hat{n} \cdot \|lkv\|)$$ (A4),

where $\hat{n} \cdot \|lkv\|$ is the dot product of the surface normal calculated from a DEM and the magnitude of the UAVSAR-provided look vector (containing the east, north, and up components).

Relative permittivity describes the ratio of the dielectric permittivity of a material to the dielectric permittivity of free space (Daniels, 2004). In dry snow, relative permittivity is determined primarily by the snow density, whereas liquid water content becomes the defining variable in wet snow (Bonnell et al., 2021; Koch et al., 2014). We concluded that the snowpack was dry throughout our field campaigns (Section 4.1). We calculated relative permittivity from the Kovacs et al. (1995) equation, which was found to have a RMSE = 54 kg m$^{-3}$ for densities derived in Colorado (Bonnell et al., 2023). The equation,

$$\varepsilon_s = \left(1 + 0.845\frac{\rho_s}{1000}\right)^2 \tag{A5},$$

calculates the relative permittivity of snow from the snow density ($\rho_s$) in kg m$^{-3}$ and represents the median of published dry snow relative permittivity equations (Di Paolo et al., 2020). We estimated the relative permittivity of the snowpack surface using an estimate of the snowpack surface density. The change in snow depth ($\Delta d_s$) is given as

$$\Delta d_s = -\frac{\lambda \varphi_s}{4\pi} \times \frac{1}{\cos\theta_{inc} - \sqrt{\varepsilon_s - \sin^2\theta_{inc}}} \tag{A6}.$$

At the UAVSAR wavelength and for a given $\theta_{inc}$ = 1.2 radians and a snow surface $\varepsilon_s$ = 1.270 ($\rho_s$ = 150 kg m$^{-3}$), phase wrapping occurs at $\Delta d_s$ = 0.72 m, or $\Delta$SWE = 108 mm. Finally, $\Delta$SWE is calculated by multiplying the snow depth by the surface density:

$$SWE = d_s \times \rho_s \tag{A7}.$$

### A.2.4 Evaluation of the Leinss et al. (2015) linear approximation for InSAR $\Delta$SWE retrievals

For dry snow, InSAR phase change has a near-linear dependence upon the change in SWE (Guneriussen et al., 2001; Leinss et al., 2015; Oveisgharan et al., 2024), and such a relation can be leveraged to derive InSAR $\Delta$SWE independent of density or relative permittivity measurements. In our study, we calculated $\Delta$SWE using the density-dependent method (Equation A6–A7) because surface density was a target variable during the surveys (Figure 3) and several previous studies have opted to use the density-dependent method because airborne platforms yield a much larger range of incidence angles than satellite platforms (e.g., Hoppinen et al., 2024; Marshall et al., 2021; Nagler et al., 2022; Tarricone et al., 2023). We evaluated the utility of the Leinss et al. (2015) approximation for $\Delta$SWE using the 16–22 March 2021 HH InSAR pair. The equation,

$$\Delta SWE = \frac{\varphi_s \lambda}{2\pi\alpha}\left(1.59 + \theta_{inc}^{\frac{5}{2}}\right)^{-1} \tag{A8},$$

modifies Eq. A6 using the Matzler (1996) permittivity model such that $\Delta$SWE is calculated from the phase change, the radar wavelength, the incidence angle, and an optimization parameter ($\alpha$). Readers are referred to Leinss et al. (2015) for a review of the optimization parameter. Given the range of incidence angles and snow densities at our field sites, we chose $\alpha$ = 1.02. The linear approximation results in nearly identical $\Delta$SWE retrievals (r=0.99; Figure S4a–c) and the comparison with GPR $\Delta$SWE retrievals yields nearly identical statistical distributions and performance statistics (Figure S4d–f). We conclude that the Leinss et al. (2015) approximation may be an appropriate alternative for $\Delta$SWE retrievals from airborne platforms.

### A.2.5 Incidence angle analysis

The incidence angles used to calculate $\Delta$SWE from the UAVSAR unwrapped phase datasets were derived by down-sampling from the Copernicus 30 m DEM. The Copernicus 30 m DEM was derived from TanDEM-X acquisitions, which operates at 9.6 GHz center frequency and the DEM has increased uncertainty over forested landscapes. Here, we evaluated the uncertainty for $\Delta$SWE retrievals caused by errors in the Copernicus-derived incidence angles by calculating incidence

angles from a 0.5 m lidar digital terrain model collected in September 2021 (Adebisi et al., 2022) over a subset of the UAVSAR swath that includes our field sites. Although the Copernicus-derived incidence angles display similar trends compared to the lidar-derived incidence angles, a comparison between the two products reveals high variability between the two products ($r = 0.08$, RMSE = 20°; Figure S5a–c). ΔSWE retrieval uncertainty was estimated through a Monte Carlo

simulation with 100 000 realizations around a mean incidence angle of 52.8° and a 20° standard deviation, approximated from the RMSE of the Copernicus-derived incidence angles (Figure S5d). A density of 150 kg m$^{-3}$ and phase change of $0.5\pi$ were used for the ΔSWE inversion (Figure S5e). From the standard deviation of simulated ΔSWE, we estimate a ΔSWE retrieval uncertainty of ±7 mm that can be attributed to the use of the Copernicus-derived incidence angles.

**A.3 GPR for SWE retrievals**

GPR is a geophysical method for subsurface imaging that, when set up in the common-offset configuration, can measure the *twt* from the antennas to a reflector of interest. We used a L-band GPR with a 1.0 GHz center-frequency and a 1.0 GHz bandwidth. GPR is a well-validated tool for estimating spatially distributed snow depth and SWE (Koh et al., 1996; Lundberg et al., 2006; McGrath et al., 2019). GPR surveys aggregate signal traces to form radargrams, which map reflection amplitudes with corresponding *twt*. For SWE retrievals, the reflector of interest is the snow-ground interface, which

manifests as the highest magnitude reflector at depth, due to the high contrast between snow and soil permittivity. The radar velocity ($v_s$) of the snowpack can be estimated from the snowpack relative permittivity (Equation A5),

$$v_s = \frac{c}{\sqrt{\varepsilon_s}} \tag{A9},$$

where $c$ is the velocity of electromagnetic waves in free space (Daniels, 2004). Then, the *twt* of the ground reflector can be converted to snow depth,

$$d_s = v_s \frac{twt}{2} \tag{A10},$$

which is subsequently converted to SWE (Equation A7).

*Data availability.* Upon acceptance, UAVSAR ΔSWE products will be archived with Dryad. GPR datasets used in this analysis are archived with the NSIDC (Bonnell et al., 2022; McGrath et al., 2021). Snow pits from the 2020 season are archived at the NSIDC (Mason

et al., 2023), while snow pits from 2021 and probed snow depths from both seasons are under review at the NSIDC. SNOTEL station data is publicly available from the NRCS and was used for the following stations: Joe Wright (https://wcc.sc.egov.usda.gov/nwcc/site?sitenum=551), Lake Irene (https://wcc.sc.egov.usda.gov/nwcc/site?sitenum=565), Willow Park (https://wcc.sc.egov.usda.gov/nwcc/site?sitenum=870), and Phantom Valley (https://wcc.sc.egov.usda.gov/nwcc/site?sitenum=688). CPWS weather station data is archived at HydroShare (Kampf et al., 2022). TLS point clouds are available at UNAVCO Inc. (Williams,

2021). NASA UAVSAR datasets are available from UAVSAR or the ASF DAAC, including InSAR pair products (i.e., unwrapped phase, coherence) and SLC products (i.e., look vectors; NASA UAVSAR, 2020, 2021). The Copernicus 30 m DEM, Copernicus Global 100 m Land Cover Dataset, and Sentinel-2 Level 2A imagery were accessed via Google Earth Engine (Gorelick et al., 2017).

*Author contributions.* Conceptualization: H.P.M., D.M., R.B. Data Curation: R.B., D.M., K.W., Y.L., Y.Z., M.S., S.K. Formal Analysis: R.B. Funding Acquisition: R.B., D.M., H.P.M. Investigation: R.B., D.M., Y.L., Y.Z., L.Z., A.O.M., E.B., C.D., K.W., M.S., S.K. Methodology: R.B., D.M., J.T., H.P.M. Visualization: R.B. Writing - Original Draft Preparation: R.B. Writing - Review & Editing: R.B., D.M., J.T., H.P.M., S.K., K.W., M.S., E.B., C.D., Y.L.

*Competing interests.* The authors declare that they have no competing interests.

*Acknowledgements.* R. Bonnell acknowledges NASA FINESST award 80NSSC20K1624 for field work and data analysis support. R. Bonnell and D. McGrath acknowledge NASA THP award 80NSSC18K1405 for field work support. A portion of data analysis was performed using the RMACC Summit supercomputer, which is supported by the NSF (awards ACI-1532235 and ACI-1532236), the University of Colorado-Boulder, and Colorado State University. We thank the NASA SnowEx leadership team for designing and implementing the 2020 and 2021 Time Series Campaigns. We thank the NSIDC and M. Mason for data archiving support. We thank the NASA UAVSAR team for their data collection and processing efforts. We acknowledge the services provided by the GAGE Facility, operated by UNAVCO, Inc., with support from the NSF and NASA under NSF Cooperative Agreement EAR-1724794. We thank Dr. E. Greene and the CAIC for sharing the CAIC Cameron Pass weather station data. We acknowledge W. Reis, C. Kane, J. C. Suhr, B. Auer, J. Nelson, and K. Snelling for their assistance with fieldwork. We thank J. Dozier, an anonymous reviewer, and G. Medici for their insightful comments and constructive feedback provided during the open review process through EGUsphere that led to an improved manuscript.

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
