# Peer review of "Evaluating L-band InSAR Snow Water Equivalent Retrievals with Repeat Ground- Penetrating Radar and Terrestrial Lidar Surveys in Northern Colorado"

_EGUsphere, 2024_

## Referee Comment (RC1)

This paper presents interesting results that offer both hope and caution for L-band InSAR and InSAR generally. The quality and extent of the field validation data are impressive, perhaps the best I've seen from a field snow experiment.

Three suggestions to improve the analysis and the presentation:

1. My major critique is that the target audience has to know a lot about radar and radar remote sensing of snow properties to understand the paper's implications. Craig Bohren has a pointed phrase about a subject being "well known to those that know it well," and this paper unfortunately hits that spot especially in the Introduction. Some colleagues tell me that the coded vocabulary makes the radar remote sensing literature hard to penetrate.
2. The approach for all the methods for SWE retrieval seems to combine a measurement of depth by some remotely sensed method, and then to multiply those depths by estimates of density. With snow depths retrieved from lidar or photogrammetry, this is the viable approach, but from InSAR data it's feasible (and probably better) to directly retrieve SWE without estimating depth or density.
3. The explanations for getting SWE from InSAR are scattered throughout: in the Introduction, Section 3 (Methods), or the Appendix. Perhaps consolidating might be the answer, or advise some readers to read the Appendix first.

Some line-by-line comments, but consider in the context of the three points above.

Line 30: maybe insert a short parenthetical definition of L-band (frequency 1-2 GHz).

Line 35: I tend to avoid adjectives ("high" here) to describe statistical measures like correlation. In some spectroscopic retrievals I work on, r<0.9 is awful. Present the values themselves.

Line 34: Is "coherence" the same as "correlation"? Without knowing that, some of the rest of the Abstract is hard to interpret. In general, this issue pervades the paper. Coherence is shown to be important but isn't defined.

Line 36: poor in one year, good the next. Any explanation? I see on Line 420 that this may be an artifact of mis-registration between airborne and in situ data.

Line 37: The sentence "We found that …" seems incongruous with RMSE between 19-22 mm. It would also be useful to specify the ranges of SWE (total) and ΔSWE (between passes) in the experiment. This information does show up later in the paper.

Line 47: The Wrzesien 2018 paper covered North America but the sentence is global. Maybe cite the 2019 paper instead (DO10.1029/2019WR025350I) or clarify that the sentence applies to North America in the 2018 paper.

Line 54: SNOTEL stations are all on nearly flat terrain, hence interpolating between them misses effects of slope and orientation. This sampling bias, combined with the spatial and elevational extent of the snow pillow network, subjects interpolation to artifacts.

Line 56, let's correct a misunderstanding: National Academies of Sciences, Engineering and Medicine are NOT a "government agency."

Line 60: I don't think SnowEx was a "mission." The Durand et al. 2018 reference uses "campaign."

Line 65: "is" not "are".

Line 72: Need a short tutorial here explaining what backscatter, time-of-flight, and co-polar phase difference are. And then a sentence about why the paper focuses on InSAR (which indeed is defensible). The reference to Borah et al. 2023 perhaps distracts. If indeed we can measure SWE up to 800 mm based on backscattering at X- and Ku-band, why go to interferometry? Earlier work by Jiancheng Shi also got impressive results based on multifrequency multipolarization backscatter, albeit with validation by a only few snow pits.
Consider this comment in the context of data processing. Then the details of how you measure coherence, time delay, phase angle, etc. (now Lines 85-107) can be covered in Section 3 or in the Appendix (but make the forward reference).

Line 74 et seq. At the first introduction of "frequency," it would be useful to include a short table that translates between "Q"-band, frequency, and wavelength. I hope that this paper will be read by people who have no idea what X-band is, or whether X-band's frequency is greater than or less than P-band's.

Line 85: Maybe a sketch here to explain what a phase change and a coherent reflection are, or cite where one can find an explanation, or refer to Section 3 or the Appendix. In the current version, it's difficult to figure out how one goes from measurement to estimate of phase change.

Line 86: "The technique was first established at C-band . . ." First established to do what? Does this remark refer especially to snow, or to interferometric retrievals of elevation?

Line 88: "interferogram" indeed well known to a small community, possibly obtuse to other readers.

Line 98: Not sure what "only two of these studies have not considered atmospheric signal delays" means. Does it imply that signal delays are important, but seemingly well covered?

Line 100-108: This paragraph has information, but not enough to know how one gets a measurement of phase difference between an interferometric pair. Also, is coherence the same as a product-moment correlation? Or something related but different?

Line 170: I suggest expanding section 3.1 with material from the Introduction (line 85-107) For the less informed reader, the relationship between coherence and phase is arcane. In particular, the snow properties that degrade coherence are important and affect the need for frequent image acquisition. How is the interferometric phase angle determined from the correlated (cohered?) pairs?

Line 177: And then we have to worry about "phase unwrapping," but this text doesn't tell us what that is. Also, is phase unwrapping a problem generally with SAR at L-band and higher frequency? Perhaps interpret the equations in Leinss et al. 2015 to explain? (Later I see phase unwrapping at ~100 mm)

Figure 2 and Line 196: Calculations of Incidence Angles from the Copernicus DEM lead to an uncertainty in cosine(incidence) of ~0.1 (from my own work, DOI 10.1029/2022JG007147), but are you able to overcome this problem because repeated images get you the right incidence geometry? Otherwise this is a source of uncertainty, even with the best available global DEM.

Line 215: Can you include a equation that defines Coherence? Or is it just Pearson product-moment correlation?

Line 235: Maybe include a citation to Reflex W? I may not need to know what a "de-wow" filter is, but I'd like to know that I could find out.

Line 248: The title of Section 3.2.3 is "TLS" but the section also covers the UAV lidar.

Line 283: "phase cycle" appears here for the first time. The cognoscenti know what this is but some readers may not.

Line 424: "phase unwrapping" is mentioned here and elsewhere. In processing the interferometric phase values, how do you decide when you've gone through a phase cycle? Or more than one?

ESTIMATING SWE DIRECTLY FROM InSAR (instead of estimating depth and multiplying by density)

Rearrange Eq. (A5) to calculate $\varphi_s$ (similar to how Leinss et al. 2015 explain):

$$\varphi_s = \frac{4\pi\Delta d_s}{\lambda}\left(-\cos\theta_i + \sqrt{\varepsilon_s - \sin^2\theta_i}\right)$$

By inspection, two snow terms drive $\varphi_s$ to increase, $\varepsilon_s$ which depends on density $\rho_s$, and $\Delta d_s$. The relationship is nearly linear, certainly linear in $\Delta d_s$ and nearly linear in $\rho_s$. $\Delta SWE = \Delta d_s \rho_s$, so different combinations of $\Delta d_s$ and $\rho_s$ can yield the same $\Delta SWE$. $\Delta SWE = f(\varphi_s)$ is nearly linear with a weak dependence on density only at combinations of deep snow with low densities.

[Figure]

Thus, a compelling argument for InSAR is its lack of dependence on density, in contrast to lidar for example where the biggest uncertainty is that in density.

---

## Author Comment (AC1)

**Response to Community Comment 1 – Dr. Giacomo Medici**

Dear Dr. Giacomo Medici,

We thank you for taking the time to review our work and provide helpful and supportive comments. We have addressed your comments below in blue. As more progress is made towards global snowpack monitoring, we hope that better connections are built between the fields of snow hydrology and hydrogeology. We believe your comments led to a better referenced and more representative introduction and a stronger conclusion.

Sincerely,

Randall Bonnell, on behalf of co-authors

General comments

Paper of large impact that can be highly cited in the future. Indeed, lots of research is focusing these days on remote sensing and the role of the snowpack in hydrology. See below my specific comments for the Discussion.

Specific comments

Abstract

1. Lines 30-40. Specify in the abstract the spatial scale. How much is the area large? The idea is to provide the observation scale in the abstract to enhance clarity.

We agree, a sense of spatial scale is critical for placing our work in context. We have revised lines 31 and 33 to include the approximate size of the UAVSAR swath and the approximate sizes of the two field sites.

Introduction

1. Lines 42. "In snow-dominated watersheds, melt from seasonal snowpacks drives streamflow and groundwater recharge". Add recent papers that show the importance of snowmelt on aquifer recharge in snow-dominated watersheds worldwide:

- Tracking flowpaths in a complex karst system through tracer test and hydrogeochemical monitoring: Implications for groundwater protection (Gran Sasso, Italy). Heliyon, https://doi.org/10.1016/j.heliyon.2024.e24663

- Snowpack aging, water isotope evolution, and runoff isotope signals, Palouse Range, Idaho, USA. Hydrology, 9(6), 94, https://doi.org/10.3390/hydrology9060094

Thank you for your suggestion! The reference that we cited (Li et al., 2017) primarily discusses the contribution of snowmelt to streamflow. We agree that including a reference to snowmelt recharge of groundwater would provide better support for our statement. Of the two references suggested, we have chosen to cite Lorenzi et al. (2024) because their findings directly tie groundwater flow to the snowmelt season.

1. Line 50. Summarize the scenario of snow decline in other mountain belts. What about Andes? See below:

- Rapid decline of snow and ice in the tropical Andes–Impacts, uncertainties and challenges ahead. Earth-science Reviews, 176, 195-213, https://doi.org/10.1016/j.earscirev.2017.09.019.

Unfortunately, snowpack development in the Andes is largely limited to glacierized basins and monitoring of snowpacks in the Andes has been extremely limited. Thus, most estimates of changes to Andean snowpacks are highly uncertain. Monitoring of Andean snowpacks is one potential application of the InSAR method that we evaluate in our manuscript.

We have opted to include snowfall projections for the Himalayas as described by Viste & Sorteberg (2015) in line 51.

Line 116. I suggest adding the three to four specific objectives of the research by using numbers (e.g., i, ii, iii).

Thank you for this suggestion. We have revised lines 112–117 to include the number of specific research objectives and to better delineate the sequence of the objectives.

Methods.

1. Line 174. "Along a ~40–60 km stretch with a 16 km swath width". I suggest inserting the link with Figure 1a.

We have revised this line to include a reference to Figure 1a and a citation of the UAVSAR dataset.

2. Line 221. I suggest "key in situ measurements included in this research are:..."

   Accepted.

3. Line 271. "3×3 pixel grid". Please, specify the size.

   Done.

Results

1. Line 305. "RMSE", specify the acronym "Root Mean Squared Error" earlier in the manuscript.

   Thank you for catching this. We have revised the first mention of RMSE in the main manuscript (Line 219) to spell out root mean squared error.

Discussion and Conclusion

1. Lines 503-518. Recall the wider implications of your paper that are part of your discussion. Please, do not simply summarize your results in the Conclusion.

   We have adjusted our conclusion to emphasize the utility of the InSAR SWE retrieval methodology as observed in our manuscript. In particular, we described the technique as having promisingly accurate statistical distributions over larger spatial scales, but low correlation coefficients for single InSAR pairs, which suggests caution for SWE interpretation at the single-pixel scale. We highlighted that our manuscript emphasized open study areas and that L-band InSAR applicability below forest cover remains an active question. Finally, we connected our automated station study to the utility of cumulative InSAR SWE retrievals, a requirement for any SWE remote sensing method.

Figures and tables

1. Figure 2. I suggest to make either the boxes for GPR and TLS workflows lighter. The green in the GPR Workflow is too dark for letters in black.

   Thank you for this suggestion. The GPR box color has been changed to a pastel red color.

2. Figure 8. Not very conceptual. Is it necessary in the manuscript? Possible to insert in the Supplementary Material?

We believe that Figure 8 represents a central finding of our study. Coherence is a measure of the similarity between two radar signals. For reference, two radar signals that are perfectly in-phase and have the same frequency and amplitude characteristics have a coherence = 1.0. Coherence degrades as one or both signals moves out-of-phase with the other. Low coherence has been presented as a primary obstacle for using InSAR for snowpack monitoring (e.g., Deeb et al., 2011). We review the processes that cause coherence degradation in the Introduction. Notably, coherence is expected to degrade with increased temporal baselines, which is a primary concern for L-band InSAR ΔSWE retrievals from the upcoming NISAR satellite mission, because NISAR will have a 12-day temporal baseline.

Figure 8a shows that L-band InSAR ΔSWE retrievals can be reliably retrieved even for lower coherences (coherence <0.4), while Figure 8b shows that moderate coherence levels are maintained for the 15-day temporal baseline pair. Thus, Figure 8 supports the application of L-band InSAR ΔSWE retrievals from the NISAR satellite and other upcoming L-band SAR satellites (e.g., the ROSE-L satellite).

References

1. Lines 644-870. Add the relevant references suggested above on the importance of the snowpack in the hydrological cycle.

Done.

**References**

Deeb, E. J., Forster, R. R., and Kane, D. L.: Monitoring snowpack evolution using interferometric synthetic aperture radar on the North Slope of Alaska, USA, International Journal of Remote Sensing, 32, 3985–4003, https://doi.org/10.1080/01431161003801351, 2011.

Li, D., Wrzesien, M. L., Durand, M., Adam, J., and Lettenmaier, D. P. (2017). How much runoff originates as snow in the western United States, and how will that change in the future?, Geophysical Research Letters, 44, 6163-6172, https://doi.org/10.1002/2017GL073551, 2017.

Lorenzi, V., Banzato, F., Barberio, M. D., Goeppert, N., Goldscheider, N., Gori, F., Lacchini, A., Manetta, M., Medici, G., Rusi, S., and Petitta, M.: Tracking flowpaths in a complex karst system through tracer test and hydrogeochemical monitoring: Implications for groundwater protection (Gran Sasso, Italy), Heliyon, 10, e24663, https://doi.org/10.1016/j.heliyon.2024.e24663, 2024.

Viste, E., and Sorteberg, A.: Snowfall in the Himalayas: an uncertain future from a little-known past, The Cryosphere, 9, 1147-1167, https://doi.org/10.5194/tc-9-1147-2015, 2015.

---

## Author Comment (AC2)

**Response to Reviewer 1**

Dear Dr. Jeff Dozier,

Thank you for your thorough review of our manuscript. We found your suggestions to be helpful and we believe your commentary has led to an improved manuscript. Specifically, we have improved the accessibility of the manuscript, reorganized and added sections for radar methodology, explored a ΔSWE retrieval method independent of density (Leinss et al., 2015), and evaluated the accuracy of the Copernicus-derived incidence angles. We addressed the manuscript's accessibility by including definitions and descriptions for many of the complex terminology and referring readers to previous studies that include sketches and more detailed descriptions of the methodology. We have added sentences to the introduction and to the appendix to improve our explanations of phase and coherence. Finally, two sections were added to the appendix to include the evaluation of ΔSWE retrieval uncertainty caused by Copernicus-derived incidence angles and a study on the Leinss et al. (2015) ΔSWE retrieval methodology. Below, you will find our responses to your review in blue. Thank you for the time you put into writing such an insightful review.

Sincerely,

Randall Bonnell, on behalf of co-authors

This paper presents interesting results that offer both hope and caution for L-band InSAR and InSAR generally. The quality and extent of the field validation data are impressive, perhaps the best I've seen from a field snow experiment.

Three suggestions to improve the analysis and the presentation:

1. My major critique is that the target audience has to know a lot about radar and radar remote sensing of snow properties to understand the paper's implications. Craig Bohren has a pointed phrase about a subject being "well known to those that know it well," and this paper unfortunately hits that spot especially in the Introduction. Some colleagues tell me that the coded vocabulary makes the radar remote sensing literature hard to penetrate.

Thank you for your candor. We feel that our edits have made this a more approachable paper without becoming too much of a review. In summary, we have added more details for radar

terminology and radar wavelengths/frequencies and we added an equation for coherence to the Appendix. Further details are provided for each of your comments below.

2. The approach for all the methods for SWE retrieval seems to combine a measurement of depth by some remotely sensed method, and then to multiply those depths by estimates of density. With snow depths retrieved from lidar or photogrammetry, this is the viable approach, but from InSAR data it's feasible (and probably better) to directly retrieve SWE without estimating depth or density.

Our approach for ΔSWE retrievals was deliberately chosen because surface density was a target observation during our field campaigns, we wanted our algorithm to be adaptable in case we found evidence for liquid water content, and there has been precedent set in recent years to use the density-dependent equation for airborne platforms (e.g., Hoppinen et al., 2024; Marshall et al., 2021; Nagler et al., 2022; Tarricone et al., 2023). The linear approximation between phase-change and SWE-change is an aspect that makes the InSAR method so appealing for large-scale SWE retrievals. We decided to maintain our focus on the density-dependent equation, but have used the Leinss et al. (2015) approximation to calculate ΔSWE retrievals for the 16–22 March 2021 HH pair for comparison with our ΔSWE retrievals. A summary of our results can be found at the end of this document, and we have added a section to the Appendix to discuss the analysis. We found the Leinss et al. (2015) approximation to be nearly identical to our density-dependent inversion and we plan to use the Leinss et al. (2015) approximation for future InSAR SWE retrieval studies.

3. The explanations for getting SWE from InSAR are scattered throughout: in the Introduction, Section 3 (Methods), or the Appendix. Perhaps consolidating might be the answer, or advise some readers to read the Appendix first.

Our intent was to give a high-level introduction to InSAR for SWE retrievals in Section 1, give a summary of the UAVSAR methods in Section 3.1, and detailed methodology in Appendix A.2. We have added a sentence in the introduction (Line 96) and at the beginning of Section 3.1 to direct readers to the appendix.

Some line-by-line comments, but consider in the context of the three points above.

Line 30: maybe insert a short parenthetical definition of L-band (frequency 1-2 GHz).

We have added your suggestion and have also added the approximate L-band wavelength.

Line 35: I tend to avoid adjectives ("high" here) to describe statistical measures like correlation. In some spectroscopic retrievals I work on, r<0.9 is awful. Present the values themselves.

We agree with your point and have adjusted the sentence accordingly.

Line 34: Is "coherence" the same as "correlation"? Without knowing that, some of the rest of the Abstract is hard to interpret. In general, this issue pervades the paper. Coherence is shown to be important but isn't defined.

Coherence and correlation are often used synonymously for InSAR. For example, the calculation of coherence is described in *Introduction to Microwave Remote Sensing* (Woodhouse, 2017) as using the complex correlation approach for interferometric radiometry. The primary difference is that radar interferometric coherence uses a spatial average whereas interferometric radiometry correlation uses a temporal average. We have specified in the abstract and the introduction that coherence refers to the complex interferometric coherence and have moved the definition of coherence to line 97.

Line 36: poor in one year, good the next. Any explanation? I see on Line 420 that this may be an artifact of mis-registration between airborne and in situ data.

Thank you for this comment. Upon review of the abstract, this sentence is an oversimplification of what we found. Yes, spatial baselines were better aligned in 2021, but poor agreement for several sequences of three InSAR pairs is observed in Figure 10a–g. For example the sequence of 2021 InSAR pairs for 15–20 January, 20–27 January, and 27 January to 3 February yielded poor agreement with the SNOTEL SWE.

We have revised this sentence to more accurately describe our results: "UAVSAR ΔSWE showed some scatter with ΔSWE measured at automated stations for both study years, but cumulative UAVSAR SWE yielded a r = 0.92 and RMSE = 42 mm when compared to total SWE measured by the stations."

Line 37: The sentence "We found that …" seems incongruous with RMSE between 19-22 mm. It would also be useful to specify the ranges of SWE (total) and ΔSWE (between passes) in the experiment. This information does show up later in the paper.

We agree and have revised this sentence to read: "Further, UAVSAR ΔSWE RMSE ranged by <10 mm from coherences of 0.10 to 0.90, suggesting that coherence has only a small influence on the ΔSWE retrieval accuracy."

We have revised lines 30–31 to state the approximate SWE accumulation that occurred during the UAVSAR campaigns.

Line 47: The Wrzesien 2018 paper covered North America but the sentence is global. Maybe cite the 2019 paper instead (DO10.1029/2019WR025350I) or clarify that the sentence applies to North America in the 2018 paper.

Thank you for catching this. We have updated the citation to Wrzesien et al. (2019)

Line 54: SNOTEL stations are all on nearly flat terrain, hence interpolating between them misses effects of slope and orientation. This sampling bias, combined with the spatial and elevational extent of the snow pillow network, subjects interpolation to artifacts.

Thank you for this suggestion. We have specified that the spatial variability of snow, coupled with the location bias and limited elevational extent of automated stations, limits interpolation.

Line 56, let's correct a misunderstanding: National Academies of Sciences, Engineering and Medicine are NOT a "government agency."

Thank you for this clarification. We have specified that snowpack monitoring via remote sensing has been set as a high priority by the National Academies of Sciences, Engineering, and Medicine.

Line 60: I don't think SnowEx was a "mission." The Durand et al. 2018 reference uses "campaign."

Accepted

Line 65: "is" not "are".

Accepted. Thank you for catching this.

Line 72: Need a short tutorial here explaining what backscatter, time-of-flight, and co-polar phase difference are. And then a sentence about why the paper focuses on InSAR (which indeed is defensible). The reference to Borah et al. 2023 perhaps distracts. If indeed we can measure SWE up to 800 mm based on backscattering at X- and Ku-band, why go to interferometry? Earlier work by Jiancheng Shi also got impressive results based on multifrequency multipolarization backscatter, albeit with validation by a only few snow pits.

 Consider this comment in the context of data processing. Then the details of how you measure coherence, time delay, phase angle, etc. (now Lines 85-107) can be covered in Section 3 or in the Appendix (but make the forward reference).

Thank you for the constructive feedback! We have provided brief explanations of SAR backscatter physics (line 75) and what time-of-flight means in the context of radar (line 72). We mention co-polar phase difference for completeness, and thus have directed readers to Leinss et al. (2014) and Patil et al. (2020).

Regarding SAR backscatter methods, we have adjusted the reference to Borah et al. (2023) to read, "...with the potential for retrieving SWE in deeper snowpacks (Borah et al., 2023)." We

have added a line regarding the snow depth retrievals from backscatter methods of Shi and Dozier (2000). Finally, we emphasized the advantage of InSAR for SWE retrievals, particularly the linear approximation between changes in SWE and phase change.

Line 74 et seq. At the first introduction of "frequency," it would be useful to include a short table that translates between "Q"-band, frequency, and wavelength. I hope that this paper will be read by people who have no idea what X-band is, or whether X-band's frequency is greater than or less than P-band's.

Thank you for this suggestion. We agree that radar vernacular can be a significant barrier for readers and it would be quite unfortunate for a reader to give up on our paper because of an accessibility issue. We have added frequency ranges and approximate wavelengths to all first mentions of radar bands.

Line 85: Maybe a sketch here to explain what a phase change and a coherent reflection are, or cite where one can find an explanation, or refer to Section 3 or the Appendix. In the current version, it's difficult to figure out how one goes from measurement to estimate of phase change.

We have added a referral to Appendix A.2. Additionally, Guneriussen et al. (2001) was provided as a citation and has a complete review of the method along with sketches that illustrate how the phase change is used to estimate a change in SWE. Thus, we feel that including a sketch here would be somewhat redundant and out of the scope of our paper.

Line 86: "The technique was first established at C-band . . ." First established to do what? Does this remark refer especially to snow, or to interferometric retrievals of elevation?

Thank you for catching this. Guneriussen et al. (2001) was the first study to use InSAR to retrieve SWE. We have clarified this sentence.

Line 88: "interferogram" indeed well known to a small community, possibly obtuse to other readers.

We have revised this line to include the definition of interferograms.

Line 98: Not sure what "only two of these studies have not considered atmospheric signal delays" means. Does it imply that signal delays are important, but seemingly well covered?

We have removed the word, "not", as this was a typo. We have also corrected "two of these studies" to "three of these studies", which includes Hoppinen et al. (2024), Oveisgharan et al. (2024), and Tarricone et al. (2023). We have also added text in line 99 to describe the cause of atmospheric delays.

Line 100-108: This paragraph has information, but not enough to know how one gets a measurement of phase difference between an interferometric pair. Also, is coherence the same as a product-moment correlation? Or something related but different?

The purpose of this paragraph is to frame coherence as a parameter that may be necessary for robust InSAR ΔSWE retrievals, but few evaluations on coherence have been published with respect to ΔSWE retrieval accuracy.

A phase-change can be calculated from any two waveforms, but a phase-change is only meaningful if the two waveforms are sufficiently coherent. Thus, coherence is a measure of similarity between two waveforms and it is different from product-moment correlation.

We have pointed readers in line 82 to Appendix A.2, where we have added a sentence that includes the equation for coherence and a description for how coherence is used for InSAR.

Line 170: I suggest expanding section 3.1 with material from the Introduction (line 85-107) For the less informed reader, the relationship between coherence and phase is arcane. In particular, the snow properties that degrade coherence are important and affect the need for frequent image acquisition. How is the interferometric phase angle determined from the correlated (cohered?) pairs?

We have added a sentence to Section 3.1 that describes phase and phase cycles and we have added information to Appendix A.2 that describes how coherence is used for InSAR applications. We feel that the material in the introduction regarding coherence degradation and the need for frequent image acquisition is appropriately placed.

Line 177: And then we have to worry about "phase unwrapping," but this text doesn't tell us what that is. Also, is phase unwrapping a problem generally with SAR at L-band and higher frequency? Perhaps interpret the equations in Leinss et al. 2015 to explain? (Later I see phase unwrapping at ~100 mm)

We have revised Section 3.1 to include a description of phase cycles and unwrapping.

Admittedly, phase unwrapping is less of an issue at L-band than it is for C-band or higher frequencies because the ±π phase cycle (~±108 mm ΔSWE for UAVSAR) expands with increasing wavelength. However, phase unwrapping is dependent on coherence. As mentioned previously, we have added a sentence that discusses phase unwrapping and coherence in Appendix A.2.

Figure 2 and Line 196: Calculations of Incidence Angles from the Copernicus DEM lead to an uncertainty in cosine(incidence) of ~0.1 (from my own work, DOI 10.1029/2022JG007147), but are you able to overcome this problem because repeated images get you the right incidence geometry? Otherwise this is a source of uncertainty, even with the best available global DEM.

Thank you for this comment. UAVSAR provides a look vector data product that includes the spatially distributed east, north, and up components of the radar signal path. This look vector is calculated as the average for the flight path and is used in conjunction with the DEM to calculate incidence angles (Appendix A.2).

We took this comment as an opportunity to better understand the uncertainty here. Two lidar flights were flown over a portion of the UAVSAR swath and both field sites in 2021. From these flights, digital elevation models have been made publicly available (Adebisi et al., 2022). We evaluated incidence angles derived from the Copernicus DEM with incidence angles derived from the lidar DEM and then ran a Monte Carlo simulation using 100,000 realizations to better understand the ΔSWE error associated with the Copernicus DEM. We found that the Copernicus and lidar incidence angles yielded similar spatial patterns, but the Copernicus-derived incidence angles failed to resolve many of the fine scale features observed in the lidar-derived incidence angles (Figure S2a–b). Overall, the two sets of incidence angles yield a low Pearson's correlation coefficient (r =0.08), but the distribution is centered on the one-to-one line. Using the calculated RMSE of 20°, a phase change of 0.5π radians, and a surface density of 150 kg m$^{-3}$ within the Monte Carlo simulation, we estimate a ΔSWE uncertainty of ±7 mm from the Copernicus-derived incidence angles.

We have added a statement in Section 3.1 to direct readers to Appendix A.2 for a review of this evaluation.

[Figure]

Figure: Incidence angles derived from (a) the Copernicus 30 m DEM and (b) the 0.5 m lidar DEM. (c) Comparison between the Copernicus-derived and the lidar-derived incidence angles. Results

from the Monte Carlo simulation of (d) incidence angles and the (e) corresponding ΔSWE. The Monte Carlo simulation was based on a mean incidence angle of 52.8° with a 20° standard deviation, a snow density of 150 kg m$^{-3}$, and a phase change of 0.5π radians.

Line 215: Can you include an equation that defines Coherence? Or is it just Pearson product-moment correlation?

We have included the coherence equation in Appendix A.2.1.

Line 235: Maybe include a citation to Reflex W? I may not need to know what a "de-wow" filter is, but I'd like to know that I could find out.

Thank you for this suggestion. We added text to describe the de-wow filter as a one-dimensional filter that removes low-frequency noise and a reference to the ReflexW software (Sandmeier, 2019).

Line 248: The title of Section 3.2.3 is "TLS" but the section also covers the UAV lidar.

Yes, the USGS, in coordination with our field efforts, collected and performed all processing on the snow-off UAV-lidar dataset. We have renamed section 3.2.3 "Lidar Scans." We have not adjusted subsequent TLS headings, labels, or phrasings because all snow-on scans were collected from a terrestrial platform.

Line 283: "phase cycle" appears here for the first time. The cognoscenti know what this is but some readers may not.

We have addressed this comment by describing phase cycles and unwrapping in Section 3.1, per the previous comment regarding Line 177.

Line 424: "phase unwrapping" is mentioned here and elsewhere. In processing the interferometric phase values, how do you decide when you've gone through a phase cycle? Or more than one?

InSAR measures phase deformation within a ±π phase cycle (~±108 mm). For some accumulation events, many pixels within a scene may see a ΔSWE >108 mm, which means that phase unwrapping is necessary to calculate the "true" ΔSWE. In brevity, phase unwrapping integrates the pixel-wise phase differences across the swath to estimate the true phase deformation. In Appendix A.2.1, we cite Goldstein and Werner (1998), the algorithm that UAVSAR implements when their team unwraps interferograms.

ESTIMATING SWE DIRECTLY FROM InSAR (instead of estimating depth and multiplying by density)

Rearrange Eq. (A5) to calculate $\varphi_s$ (similar to how Leinss et al. 2015 explain):

$$\varphi_s = \frac{4\pi\Delta d_s}{\lambda}\left(-cos\theta_i + \sqrt{\varepsilon_s - sin^2\theta_i}\right)$$

By inspection, two snow terms drive $\varphi_s$ to increase, $\varepsilon_s$ which depends on density $\rho_s$, and $\Delta d_s$. The relationship is nearly linear, certainly linear in $\Delta d_s$ and nearly linear in $\rho_s$. $\Delta SWE = \Delta d_s\rho_s$, so different combinations of $\Delta d_s$ and $\rho_s$ can yield the same $\Delta SWE$. $\Delta SWE = f(\varphi_s)$ is nearly linear with a weak dependence on density only at combinations of deep snow with low densities.

[Figure]

Thus, a compelling argument for InSAR is its lack of dependence on density, in contrast to lidar for example where the biggest uncertainty is that in density.

We wholeheartedly agree with your assertion! The ability to retrieve SWE without density makes the L-band InSAR method particularly promising for global SWE monitoring applications. When we began our analysis, we specifically designed our scripts to use the density-dependent method because surface density was set as a target observation during the surveys and we wanted to use an equation that could accommodate any potential liquid water content in the snowpack. Through our analysis, we determined that liquid water content was not likely present in the snowpack at our field sites during the UAVSAR flights.

Although approximations such as the Leinss et al. (2015) equation are the most likely InSAR equations for global SWE retrievals, the density-dependent method we implemented is an appropriate and accurate approach, particularly given that the method is relatively insensitive to the input density (Hoppinen et al., 2024). Additionally, there is precedent for the density-dependent method to be used for airborne platforms, which tend to have a larger range of

incidence angles than satellite platforms, making the Leinss et al. (2015) approximation a bit more uncertain. For reference, recent airborne L-band InSAR studies that have used the density-dependent method include Hoppinen et al. (2024), Marshall et al. (2021), Nagler et al. (2022), and Tarricone et al. (2023).

We tested the Leinss et al. (2015) approximation using the 16–22 March 2021 HH InSAR pair to determine its appropriateness for the UAVSAR platform (see figure below). Scene-wide ΔSWE retrievals are identical between the density-dependent method that we implemented in the study and the Leinss et al. (2015) approximation (r = 0.99; Figure a–c). When evaluated using the GPR ΔSWE retrievals, both methods yield identical Pearson's correlation coefficients and RMSEs (Figure d–f). We have added the analysis and methods of this evaluation to Appendix 2 and the figure has been added to the supplement. We conclude that, for dry snow, the Leinss et al. (2015) method is applicable from airborne platforms. However, we have decided to maintain our focus on the density-dependent method because the central findings and implications would likely remain the same, but extensive small changes to figures and statistics throughout the manuscript would be required if we fully adopted the Leinss et al. (2015) approximation.

[Figure]

Figure: Evaluation of the Leinss et al. (2015) approximation for ΔSWE retrievals from the 16–22 March 2021 HH InSAR pair. ΔSWE retrievals calculated from (a) the density-dependent equation used in the manuscript and (b) the Leinss et al. (2015) approximation. (c) Comparison between the density-dependent and Leinss et al. (2015) approximation ΔSWE retrievals. Comparison of GPR ΔSWE retrievals with (d) the density-dependent method and (e) the Leinss et al. (2015) approximation. (f) Box plot distributions of ΔSWE retrievals from the three methods. For plots a–c, the range of ΔSWE is limited to ±75 mm, which represents >99% of the distribution.

**References**

Adebisi, H., Marshall, H., O'Neel, S., Vuyovich, C. M., Hiemstra, C., and Elder, K.: SnowEx20–21 QSI Lidar DEM 0.5 UTM Grid, Version 1, NASA National Snow and Ice Data Center Distributed Active Archive Center [data set], https://doi.org/10.5067/YO583L7ZOLOO, 2022.

Borah, F. K., Tsang, L., and Kim, E.: SWE Retrieval Algorithms Based on the Parameterized BI-Continuous DMRT Model Without Priors on Grain Size Or Scattering Albedo, Progress in Electromagnetics Research, 178, 129-147, https://doi.org/10.2528/PIER23071101, 2023.

Guneriussen, T., Hogda, K. A., Johnsen, H., and Lauknes, I.: InSAR for estimation of changes in snow water equivalent of dry snow, IEEE Transactions on Geoscience and Remote Sensing, 39, 2101–2108, https://doi.org/10.1109/36.957273, 2001.

Hoppinen, Z. M., Oveisgharan, S., Marshall, H.-P., Mower, R., Elder, K., and Vuyovich, C.: Snow Water Equivalent Retrieval Over Idaho, Part 2: Using L-band UAVSAR Repeat-Pass Interferometry, The Cryosphere, 18, 575-592, https://doi.org/10.5194/tc-18-575-2024, 2024.

Leinss, S., Parrella, G., and Hajnsek, I: Snow height determination by polarimetric phase differences in X-band SAR data. IEEE Journal of Selected Topics in Applied Earth Observations and Remote Sensing, 7, 3794-3810, https://doi.org/10.1109/JSTARS.2014.2323199, 2014.

Leinss, S., Wiesmann, A., Lemmetyinen, J., and Hajnsek, I.: Snow water equivalent of dry snow measured by differential interferometry, IEEE Journal of Selected Topics in Applied Earth Observations and Remote Sening, 8, 3773-3790, https://doi.org/10.1109/JSTARS.2015.2432031, 2015.

Marshall, H. P., Deeb, E., Forster, R., Vuyovich, C., Elder, K., Hiemstra, C., and Lund, J.: L-Band InSAR Depth Retrieval During the NASA SnowEx 2020 Campaign: Grand Mesa, Colorado, in: 2021 IEEE International Geoscience and Remote Sensing Symposium IGARSS, 2021 IEEE International Geoscience and Remote Sensing Symposium IGARSS, 625–627, https://doi.org/10.1109/IGARSS47720.2021.9553852, 2021.

Nagler, T., Rott, H., Scheiblauer, S., Libert, L., Mölg, N., Horn, R., Fischer, J., Keller, M., Moreira, A., and Kubanek, J.: Airborne Experiment on Insar Snow Mass Retrieval in Alpine Environment, in: IGARSS 2022 - 2022 IEEE International Geoscience and Remote Sensing Symposium, IGARSS 2022 - 2022 IEEE International Geoscience and Remote Sensing Symposium, 4549–4552, https://doi.org/10.1109/IGARSS46834.2022.9883809, 2022.

Oveisgharan, S., Zinke, R., Hoppinen, Z., and Marshall, H. P.: Snow Water Equivalent Retrieval Over Idaho, Part 1: Using Sentinel-1 Repeat-Pass Interferometry, The Cryosphere, 18, 559-574, https://doi.org/10.5194/tc-18-559-2024, 2024.

Patil, A., Singh, G., and Rüdiger, C.: Retrieval of Snow Depth and Snow Water Equivalent Using Dual Polarization SAR Data, Remote Sensing, 12, 1183, https://doi.org/10.3390/rs12071183, 2020.

Sandmeier, K. J.: ReflexW–GPR and Seismic Processing Software, Sandmeier [software], https://www.sandmeier-geo.de/reflexw.html, 2019.

Shi, J. and Dozier, J.: Estimation of Snow Water Equivalence Using SIR-C/X-SAR, Part II: Inferring Snow Depth and Particle Size, IEEE Transactions on Geoscience and Remote Sensing, 38, 2475–2488, https://doi.org/10.1109/36.885196, 2000.

Tarricone, J., Webb, R. W., Marshall, H.-P., Nolin, A. W., and Meyer, F. J.: Estimating snow accumulation and ablation with L-band interferometric synthetic aperture radar (InSAR), The Cryosphere, 17, 1997–2019, https://doi.org/10.5194/tc-17-1997-2023, 2023.

Woodhouse, I. H.: Introduction to Microwave Remote Sensing, CRC Press, ISBN 036722514X, 2017.

---

## Author Comment (AC3)

**Response to Reviewer 2**

Dear Reviewer 2,

We thank you for your thorough and very detailed review. We believe your suggestions have led to a more vernacularly accurate manuscript with more thorough descriptions of the radar methods. Specifically, we have added more detailed information about coherence and how it is calculated, we have added reasons for our methodological choices, and we evaluated the Leinss et al. (2015) approximation for ΔSWE retrievals with the 16–22 March 2021 HH InSAR pair. Detailed commentary is provided below in blue. Thank you so much for the time you took to provide your perspective on our work.

Sincerely,
Randall Bonnell, on behalf of co-authors

Review

General Comment

The article ‚Evaluating L-band InSAR Snow Water Equivalent Retrievals with Repeat Ground-Penetrating Radar and Terrestrial Lidar Surveys in Northern Colorado' compares SWE change retrievals from airborne interferometric SAR data, ground penetrating radar, terrestrial lidar scans, automated measurement stations and in-situ measurements. The paper provides an extensive data analysis for two winters (2020 and 2021) over different test sites in Colorado. The authors also analyze the impact of low coherence on the SWE change retrieval, providing valuable insights for future space borne L-band SAR missions.

Specific Comments

Line 37: Maybe you can add half a sentence why the agreement was poor in 2020.

Thank you for this suggestion. We feel that the statement is not representative of our results. For example, if you select any three consecutive InSAR pairs from 2021 in Figure 10, you may see poor agreement. We have revised this sentence to read, "UAVSAR ΔSWE showed some scatter with ΔSWE measured at automated stations for both study years, but cumulative UAVSAR SWE yielded a r = 0.92 and RMSE = 42 mm when compared to toal SWE measured by the stations."

Line 98: The meaning of this sentence is hard to understand. Maybe you could rephrase it.

Thank you for this suggestion, the sentence has been revised to state the cause of atmospheric delays and to correct a typo: "...only two of these studies have not considered atmospheric signal delays…" to "only three of these studies have considered atmospheric delays."

Line 100-108: You could think of adding this paragraph about the interferometric coherence to Appendix A.2., where you also describe the interferometric phase.

The purpose of this paragraph is to provide a high-level summary of coherence as an InSAR parameter and review the recent studies that have analyzed its influence upon SWE retrieval accuracy. We have decided to leave the paragraph as is, but have added more detailed information about coherence in Appendix A.2.

Line 124: Why have you used a different heading for the 27.01. /03.02. interferogram and not the 141° as well?

At the time of data analysis, the 141° heading for 27 January to 3 February 2021 was not available because of a >70 m deviation from the spatial baseline. We are aware that a product has since been calculated, but because of this significant baseline deviation, we are hesitant to incorporate it into our analysis and prefer the 321° pair because of its significantly tighter spatial baseline.

Line 177: Do you know why the coherence was low for that interferogram? Please elaborate briefly.

We are uncertain about the exact cause of the low coherence for the 10–16 March interferogram. Between those dates, a significant snow storm deposited ~70 mm of SWE, as measured by the nearby Joe Wright SNOTEL station, which is expected to reduce coherence. Another possibility is the development of ice lenses in the snowpack (noted in Section 4.1), which would alter the backscattering properties between acquisitions. A final and likely possibility is that the >3 m deviation from the spatial baseline between the two acquisitions yielded phase changes that were unaccounted for in the topographic phase correction performed by UAVSAR.

Line 183: Maybe you can point out that it is a SWE change retrieval, so you can just measure changes between the measurements, and not directly the total SWE.

Thank you for this suggestion. We have added the clarification.

Line 211: Were the 20% GPR SWE change retrievals used for estimating the absolute phase selected randomly? And why have you not used the In-Situ stations for absolute phase calibration?

Yes, the phase calibration was performed using a randomly selected set of 20% of the GPR ΔSWE Retrievals. We have added this statement to Line 211 for clarification.

We opted to use the GPR ΔSWE retrievals for phase calibration instead of weather stations for a two primary reasons:

1. Coherence at weather stations was often <0.4 (Figure 10), whereas the coherence distribution along the GPR transects was centered on ~0.65 (Figure 8a). Although we show that there is not much variability in InSAR ΔSWE error based on coherence, we did not know this going into the study and wanted to optimize the phase calibration using the measurements that had higher coherence.
2. Even for the dates where <200 UAVSAR pixels had coincident GPR measurements, we still had >25 GPR measurements to derive a statistical phase calibration. We found this to be a favorable statistical approach as compared to the, at most, seven automated stations.

Line 271: What is the resolution your 3x3 pixel grid?

The UAVSAR pixels have a spatial resolution of ~5 m x ~5 m, therefore the resolution of the 3x3 grid is ~15 m x ~15 m. We have added this to the text.

Line 352: Why have you chosen HH and not VV? Since the RMSE is smaller for VV.

Previous studies have noted that retrievals are valid from any of the polarizations, but co-polarized datasets are preferred because of the stronger strength of the backscattered signal (e.g., Palomaki & Sproles, 2023). Therefore, we aimed to present a time series from either HH or VV. Of the polarization time series, the HH polarization is only missing data from the 3–23 February 2021 InSAR pair, whereas VV is missing data from both the 12–19 February 2020 and the 3–23 February 2021 InSAR pairs. Thus, it is our opinion that the time series from the HH polarization is the easiest of the two to explain. As we state in Section 4.3, overall RMSEs are similar between all four polarizations and we do not think the 2 mm difference in overall RMSE warrants the use of a different polarization.

Line 492: In this paragraph you are discussing the influence of wet snow. Maybe you can also add that wet snow increases the absorption and decreases the penetration depth of the radar wave in the snow volume.

Thank you for this note. We have added a sentence about the effects of liquid water content upon the radar signal and described the uncertainty of the location of the amplitude-center at higher water contents.

Line 520: In your Appendix A.1 you describe the L-Band transmissibility, which is very interesting, but you never refer to A.1.

We have added a line to the introduction (Line 81) to point readers to Appendix A.1 for a review of the transmissibility of L-band radar through snow.

Line 538: (Referring to comment on line 100-108). Maybe you could also add here an equation for the interferometric coherence, so it is easier to understand what it means and how you can obtain the interferometric phase.

Thank you for this note. We agree that the appendix is a sensible location for the coherence equation, particularly since the phase unwrapping method relies upon it. The equation and a brief explanation have been added to Appendix A.2.

Line 548: In A.2.2 you are describing the atmospheric correction for UAVSAR. I am not an expert in this field, but I understood that you are estimating a phase ramp due to the atmosphere and then are checking if the atmospheric correction is improving your SWE change estimates. You stated that it does not improve your results. But where does your calculated phase ramp then come from? Maybe you could explain this more.

We appreciate your discussion of this topic. The calculated phase ramp is the result of a linear regression between the radar signal path length (i.e., distance between the SAR and the ground reflector) and the unwrapped phase of pixels that were identified as snow-free through the Normalized Difference Snow Index Analysis. The regression equation is then applied to the unwrapped interferograms to account for the atmospheric contribution to the phase delay. However, as we note in Appendix A.2.2, the estimated atmospheric corrections were generally poor and did not improve the ΔSWE retrieval accuracy compared to the automated stations.

Line 595: In (Leinss et al., 2015) an approach was presented, where a linear function between the SWE change and interferometric phase was derived. This has the advantage that you can directly derive the SWE change from the phase without the need of additional in-situ density measurements, which is the main advantage of the DInSAR approach compared to the GPR or LIDAR retrieval. Maybe you can think about it, since you also stated in line 66 that the need of in-situ density measurements adds uncertainty. Or state why you have chosen to use the approach with Equation (A5) and (A6).

Yes, we absolutely agree that the linear approximation, which is independent of density, is a primary factor that makes the L-band InSAR method for global SWE retrievals promising! We designed our scripts to use the density-dependent method because surface density was set as a target observation during the surveys and we were unsure whether liquid water content was within the snowpack during UAVSAR flights in March of 2020 and 2021. The Leinss et al. (2015) approximation was developed exclusively for dry snow applications as defined by the dry snow permittivity model that was used in their linear approximation. As we conclude in our manuscript, it is unlikely that liquid water was present at our field sites during UAVSAR flights, hence the Leinss et al. (2015) approximation may be appropriate for our analysis. However, incorporating the approximation at this stage instead of the density-dependent method that we used would require extensive changes to statistics and methods throughout the manuscript for what may result in a nearly identical evaluation. Thus, we opted to retain our focus on the density-dependent method, which is an appropriate and accurate approach, particularly given that the method is relatively insensitive to the input density (Hoppinen et al., 2024). Additionally, there is precedent for the density-dependent method to be used for airborne platforms, which tend to have a larger range of incidence angles than satellite platforms, making the Leinss et al. (2015) approximation a bit more uncertain. For reference, recent airborne L-band InSAR studies that have used the density-dependent method include Hoppinen et al. (2024), Marshall et al. (2021), Nagler et al. (2022), and Tarricone et al. (2023).

We decided to test the Leinss et al. (2015) approximation using the 16–22 March 2021 HH InSAR pair to determine its appropriateness for the UAVSAR platform (see figure below). Scene-wide ΔSWE retrievals are identical between the density-dependent method that we implemented in the study and the Leinss et al. (2015) approximation (r = 0.99; Figure a–c). When evaluated using the GPR ΔSWE retrievals, both methods yield identical Pearson's correlation coefficients and RMSEs (Figure d–f). We have added the analysis and methods of this evaluation to Appendix 2 and the figure has been added to the supplement. We conclude that, for dry snow, the Leinss et al. (2015) method is applicable from airborne platforms.

[Figure]

Figure: Evaluation of the Leinss et al. (2015) approximation for ΔSWE retrievals from the 16–22 March 2021 HH InSAR pair. ΔSWE retrievals calculated from (a) the density-dependent equation used in the manuscript and (b) the Leinss et al. (2015) approximation. (c) Comparison between the density-dependent and Leinss et al. (2015) approximation ΔSWE retrievals. Comparison of GPR ΔSWE retrievals with (d) the density-dependent method and (e) the Leinss et al. (2015) approximation. (f) Box plot distributions of ΔSWE retrievals from the three methods. For plots a–c, the range of ΔSWE is limited to ±75 mm, which represents >99% of the distribution.

Line 598: You maybe could point out that these are snow depth changes and SWE changes.

Accepted.

Technical Corrections

Line 53: Reference for the SNOTEL stations

Accepted.

Line 183: ...is outlined in Appendix A.2?

Agreed. Thank you for this suggestion.

Line 236: The GPR Workflow in Figure 2 shows the Radargram processing and it has first the step (4) and then the step (3). Maybe you can make it more consistent.

You are correct in your assertion, the trace interpolation step is performed before the 2-d filter step. Thank you for catching this typo. The GPR workflow figure has been changed to accurately reflect the processing flow.

Line 320, Line 333: It is hard to see the points in Figure 5 (i) and Figure 6 (i).

Unfortunately, phase unwrapping issues for the 3–23 February 2021 InSAR pair removed any potential evaluation at the MR field site, hence Figure 5i does not have any GPR points visible. We have added a sentence explaining this to the figure caption.

I assume that this is a reference to Figure 6f, rather than Figure 6i. This subplot represents the 3–23 February 2021 InSAR pair at the CP field site. For this interval, ΔSWE was large and maximized the color scheme, hence the difficulty identifying the GPR points. We have changed the outline of the GPR points to improve the contrast.

Line 352: Space after Figure missing.

Addressed. Thank you for catching this typo.

Line 352: Parenthesis after HH.

We believe this issue has been addressed in our response to your comment for line 353.

Line 353: In the Table S4 the RMSE is 21mm and not 22mm.

We appreciate your eye for detail. The overall RMSE for the HH polarization is 21 mm. However, as shown in Table S4, this does not include any SWE retrievals from the 3–23 February 2021 InSAR pair. We have added a sentence to Section 4.3 that specifies that, for the presented analysis, we used the HH pol for all pairs except the 3–23 February 2021 which used the VH pol. We have also added a sentence in the description for Table S4 that describes the differences between the overall statistics reported in the table and the overall statistics reported in the manuscript. We hope that this clears up the confusion.

Line 375, Line 378, Line 382: There is no Figure 8 (a-i).

Thank you for catching this mistake. The figure references have been updated to reference Figure 9 (a–i).

Supplementary Material:

Line 62:  There is only Figure 8 (a-b).

Thank you for catching this. It has been corrected.

**References**

Hoppinen, Z. M., Oveisgharan, S., Marshall, H.-P., Mower, R., Elder, K., and Vuyovich, C.: Snow Water Equivalent Retrieval Over Idaho, Part 1: Using L-band UAVSAR Repeat-Pass Interferometry, The Cryosphere, 18, 575-592, https://doi.org/10.5194/tc-18-575-2024, 2024.

Leinss, S., Wiesmann, A., Lemmetyinen, J., and Hajnsek, I.: Snow water equivalent of dry snow measured by differential interferometry, IEEE Journal of Selected Topics in Applied Earth Observations and Remote Sening, 8, 3773-3790, https://doi.org/10.1109/JSTARS.2015.2432031, 2015.

Marshall, H. P., Deeb, E., Forster, R., Vuyovich, C., Elder, K., Hiemstra, C., and Lund, J.: L-Band InSAR Depth Retrieval During the NASA SnowEx 2020 Campaign: Grand Mesa, Colorado, in: 2021 IEEE International Geoscience and Remote Sensing Symposium IGARSS, 2021 IEEE International Geoscience and Remote Sensing Symposium IGARSS, 625–627, https://doi.org/10.1109/IGARSS47720.2021.9553852, 2021.

Nagler, T., Rott, H., Scheiblauer, S., Libert, L., Mölg, N., Horn, R., Fischer, J., Keller, M., Moreira, A., and Kubanek, J.: Airborne Experiment on Insar Snow Mass Retrieval in Alpine Environment, in: IGARSS 2022 - 2022 IEEE International Geoscience and Remote Sensing Symposium, IGARSS 2022 - 2022 IEEE International Geoscience and Remote Sensing Symposium, 4549–4552, https://doi.org/10.1109/IGARSS46834.2022.9883809, 2022.

Palomaki, R. T. and Sproles, E. A.: Assessment of L-band InSAR snow estimation techniques over a shallow, heterogeneous prairie snowpack, Remote Sensing of Environment, 296, 113744, https://doi.org/10.1016/j.rse.2023.113744, 2023.

Tarricone, J., Webb, R. W., Marshall, H.-P., Nolin, A. W., and Meyer, F. J.: Estimating snow accumulation and ablation with L-band interferometric synthetic aperture radar (InSAR), The Cryosphere, 17, 1997–2019, https://doi.org/10.5194/tc-17-1997-2023, 2023.